# Common protein-coding variants influence the racing phenotype in galloping racehorse breeds

Haige Han[1], Beatrice A. McGivney [2], Lucy Allen[3], Dongyi Bai[1], Leanne R. Corduff[2], Gantulga Davaakhuu[4], Jargalsaikhan Davaasambuu[5], Dulguun Dorjgotov[6], Thomas J. Hall [7], Andrew J. Hemmings[3], Amy R. Holtby [2], Tuyatsetseg Jambal[6], Badarch Jargalsaikhan[8], Uyasakh Jargalsaikhan[5], Naveen K. Kadri [9], David E. MacHugh [7,10], Hubert Pausch [9], Carol Readhead[11], David Warburton[12], Manglai Dugarjaviin[1✉] & Emmeline W. Hill [2,7✉]

Selection for system-wide morphological, physiological, and metabolic adaptations has led to extreme athletic phenotypes among geographically diverse horse breeds. Here, we identify genes contributing to exercise adaptation in racehorses by applying genomics approaches for racing performance, an end-point athletic phenotype. Using an integrative genomics strategy to first combine population genomics results with skeletal muscle exercise and training transcriptomic data, followed by whole-genome resequencing of Asian horses, we identify protein-coding variants in genes of interest in galloping racehorse breeds (Arabian, Mongolian and Thoroughbred). A core set of genes, *G6PC2, HDAC9, KTN1, MYLK2, NTM, SLC16A1* and *SYNDIG1*, with central roles in muscle, metabolism, and neurobiology, are key drivers of the racing phenotype. Although racing potential is a multifactorial trait, the genomic architecture shaping the common athletic phenotype in horse populations bred for racing provides evidence for the influence of protein-coding variants in fundamental exercise-relevant genes. Variation in these genes may therefore be exploited for genetic improvement of horse populations towards specific types of racing.

[1] Inner Mongolia Key Laboratory of Equine Genetics, Breeding and Reproduction, College of Animal Science, Equine Research Center, Inner Mongolia Agricultural University, Hohhot 010018, China. [2] Plusvital Ltd, The Highline, Dun Laoghaire Business Park, Dublin A96 W5T3, Ireland. [3] Royal Agricultural University, Cirencester, Gloucestershire GL7 6JS, UK. [4] Institute of Biology, Mongolian Academy of Sciences, Peace Avenue 54B, Ulaanbaatar 13330, Mongolia. [5] Ajnai Sharga Horse Racing Team, Encanto Town 210-11, Ikh Mongol State Street, 26th KhorooBayanzurkh district Ulaanbaatar 13312, Mongolia. [6] School of Industrial Technology, Mongolian University of Science and Technology, Ulaanbaatar 661, Mongolia. [7] UCD School of Agriculture and Food Science, University College Dublin, Belfield, Dublin D04 V1W8, Ireland. [8] Department of Obstetrics and Gynecology, Mongolian National University of Medical Sciences, Ulaanbaatar 14210, Mongolia. [9] Animal Genomics, Institute of Agricultural Sciences, ETH Zürich, Universitätstrasse 2, 8092 Zürich, Switzerland. [10] UCD Conway Institute of Biomolecular and Biomedical Research, University College Dublin, Belfield, Dublin D04 V1W8, Ireland. [11] Biology and Bioengineering, California Institute of Technology, Pasadena, CA 91125, USA. [12] The Saban Research Institute, Children's Hospital Los Angeles, Keck School of Medicine, University of Southern California, Los Angeles, CA 90027, USA. ✉email: dmanglai@163.com; emmeline.hill@ucd.ie

Horse racing is among the oldest known sports, the general concept of which—horses, either mounted or harnessed, travelling at speed over a certain distance and terrain with the horse finishing first being the winner—has been largely unchanged for millennia[1,2]. The quest for wealth and status in racing traverses diverse cultures and geographies and today mounted horse racing is a globally popular sport. Different populations of horse have been developed for racing through a process of selective breeding for attributes required to excel in specific types of competition; these include breeds that compete in harness and trotting races, and others, including the Arabian, Mongolian and Thoroughbred, that are galloping breeds.

The Mongolian is one of the oldest extant horse populations and although domesticated, most animals are free ranging and experience minimal human intervention[3]. Mongolian horse populations have relatively high genomic diversity compared to other breeds[4,5], which may reflect the role of the central Asian steppe region as an important centre for horse domestication[6]. The Mongolian is generally classified as a breed *in toto* but several phenotypically and genetically distinct subpopulations exist[4], which are used for meat, milk, transport, and racing. Racing in Mongolia is celebrated annually during the Nadaam festival of racing where adult horses race over long distances (25–30 km) and harsh terrain. In recent years, Thoroughbred stallions have been imported for crossbreeding intended to improve speed traits in the racing populations.

The Arabian is also an ancient breed with high levels of genetic diversity[5,7]. Bred for millennia and developed by Bedouin nomads for transport and military use, the Arabian has traditionally excelled in long distance endurance racing (80–160 km), often in extreme climatic conditions. More recently there has been strong selection among subgroups for an aesthetic conformation phenotype, which is valued in show competition, and for short-distance track racing (~1600 m). Among track racing Arabians there is evidence of recent Thoroughbred crossbreeding, presumably for the introduction of speed, with some horses having up to 60% Thoroughbred ancestry[7]. Although considered a highly polygenic trait, sequence variants at several genes have been reported to be directly associated with performance traits in the Arabian breed[7–10].

Compared to the Arabian and Mongolian breeds, the Thoroughbred was developed relatively recently during the last three centuries by crossing native British and Irish mares with stallions imported from the Middle East[11]. Most Thoroughbreds compete in races over much shorter distances (1000–3200 m) on maintained track surfaces and are bred for both speed and stamina attributes[12]. Originating from a very small number of founders[13], and with subsequent restricted gene flow since the formation of the stud book[14], the Thoroughbred now has very low levels of genetic diversity despite a large global census population size[5,15,16]. These population demographics, coupled with constant human-mediated selection pressures, have resulted in athletic traits with genetic architectures that are especially amenable to modern genomics, particularly because of the high levels of linkage disequilibrium observed across the Thoroughbred genome[17] with haplotypes extending >4 Mb in regions under selection[5,12]. As a result, genome-wide association studies (GWAS) have been successfully deployed to identify quantitative trait loci (QTLs) for complex traits using relatively modest sample sizes[18–21]. Investigations of genomic targets of selection in the Thoroughbred[5,15,16,22,23], and functional analyses of gene expression profiles in skeletal muscle[24–26], have identified suites of genes and molecular pathways that are enriched for functions in energy metabolism, muscle contraction, haemostasis, organismal growth and development, lipid metabolism, the mitochondrion, fatty acid metabolism, cardiovascular signalling,

cellular stress and injury, and neurotransmitters and other nervous system signalling.

The recently developed omnigenic model proposes that phenotypic outcomes for eukaryotic complex traits are directly shaped by core genes that are embedded in highly interconnected tissue-specific gene regulatory networks, which are substantially modulated by very large numbers of genetic variants of small effect at peripheral genes across the entire genome[27]. The most well studied and arguably core exercise-relevant gene in athletic horses is the myostatin gene (*MSTN*), where a SINE insertion promoter polymorphism[28] profoundly affects skeletal muscle development[29,30] and distribution of muscle fibre types[5,31,32]. The combined effect of the major *MSTN* QTL and additive genetic variation across the genome is illustrated by recent work examining the genomic architecture and heritability of optimum race distance in multiple Thoroughbred populations[33].

In addition to myriad genetic contributions, understanding the biological basis of complex traits is further challenged by the variation in multi-dimensional system-wide endophenotypes that can be dynamic and influenced by the environment, a concept that emerged initially in neurobiological genomics[34]. Like all athletes, in addition to multiple anatomical, physiological, and metabolic processes, environmental factors and interactions also determine the elite athletic phenotype of racehorses[35,36]. Consequently, numerous endophenotypes likely contribute to biological systems relevant to the equine athlete. However, notwithstanding this complex vista, relatively straightforward selection signal detection approaches—without recourse to accurately measured endophenotypes such as hormone levels or other biomarkers—can be used to identify genes or genomic regulatory elements containing sequence variants contributing to recent evolutionary adaptation and important physiological traits in livestock species[37–40].

In the present study, we hypothesised that galloping racing breeds harbour signals of selection that contain genomic loci with sequence variants contributing to racing ability. To refine the selection signals, we assigned functional relevance to SNPs that were in proximity to differentially expressed genes (DEGs) in Thoroughbred skeletal muscle[24] and then interrogated prioritised genomic regions for putative functional protein-coding variants identified from whole genome sequence (WGS) data generated from Asian landrace horse breeds. Based on predicted variant effects and the biological functions of the genes containing them, we hypothesised that these variants may contribute to variation in racing ability among horse breeds. To further explore and validate this hypothesis, we generated genotypes for a panel of these variants in independent sample sets of Thoroughbred and Mongolian Racing horses and in other racing and athletic horse breeds, and examined whether they were associated with racing traits. The overall goal of this research was to identify a set of genetic markers contributing to athletic performance in populations of horses bred for competitive racing. These markers may be used for the improvement of racing populations, including among Mongolian horses, for which there has been less opportunity for systematic pedigree and phenotype selection that is advanced in Thoroughbred and the other more common racing breeds.

## Results and discussion

**Population structure**. An overview of the sequential steps of the study is provided in Supplementary Fig. 1. We first evaluated genetic relatedness and population structure based on genome-wide SNP-array derived genotypes among the Racing breeds (Arabian, Mongolian Racing and Thoroughbred) in the context of non-Racing breeds using a principal component analysis (PCA)

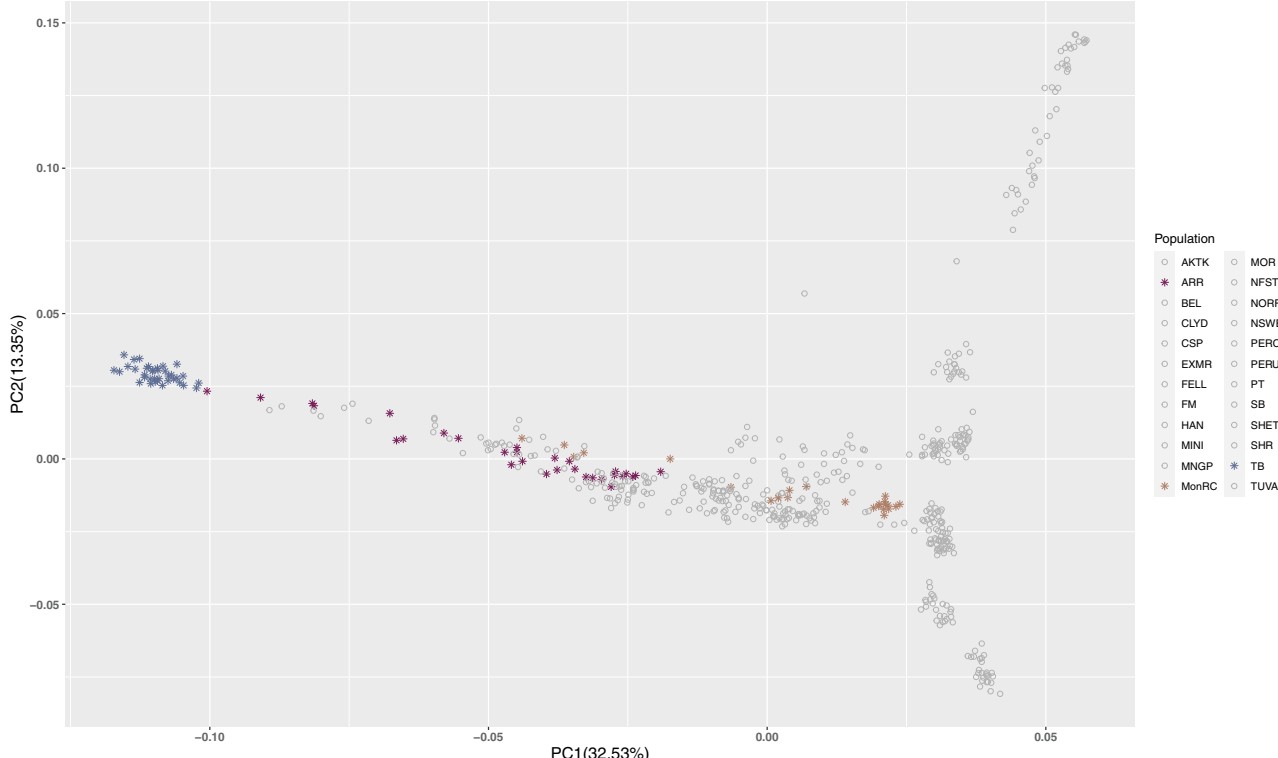

**Fig. 1 Principal component analysis (PCA) plot for $n = 574$ horses using 36,767 genome-wide SNPs.** Thoroughbreds (TB), Arabian (ARR) and Mongolian Racing (MonRC) are highlighted—TB (blue), ARR (purple), MonRC (brown).

(Fig. 1) and an admixture structure plot (Fig. 2). In the PCA plot of PC1 and PC2, Thoroughbreds were tightly clustered with no overlap with any other breed, while Arabian and Mongolian Racing horses were more loosely distributed across PC1 (32.5% of the variance) and there was some overlap between the two populations although a shared recent ancestry was not expected.

Since Thoroughbred admixture has been observed in track racing Arabians[7] and there has been an introduction of Thoroughbred stallions to Mongolia in recent years, we assessed the contribution of Thoroughbred ancestry in the Arabian and Mongolian Racing populations (Fig. 2). In the admixture structure analysis, the Belgian breed was used as a comparator population as it is distantly related to all three Racing breeds[5]. The lowest cross validation error for $K = 2$–6 modelled ancestral populations was observed for $K = 4$ (Fig. 2); therefore, this value was considered the most suitable for evaluating ancestry and quantifying admixture[41,42]. The Belgian and Thoroughbred displayed minimal evidence of admixture arising from the other breeds. Five Arabian samples exhibited >50% Thoroughbred genetic ancestry and eight had no Thoroughbred contribution. Except for five animals, the Mongolian Racing samples had some shared ancestry with the other breeds and Thoroughbred ancestry was >50% in one sample. There was minimal sharing of genetic background between the Mongolian Racing and Arabian populations. Based on the structure plot, it is clear that there is Thoroughbred ancestry in many of the Mongolian Racing and Arabian animals, and this is reflected in their position along PC1. Supplementary Data 1 details the individual ancestry contributions at $K = 4$ modelled ancestral populations for animals in the study.

Separate PCA plots were also generated for the Arabian and Mongolian Racing populations genotyped in this study compared to other Arabian horses[7] (Supplementary Fig. 2) and Mongolian horses indigenous to Inner Mongolia, China[4] (Supplementary Fig. 3). The Arabian horses were genetically diverse and were

distributed predominantly across PC2 (16.2% of the variance), which encompassed most of the Arabian variation to the exclusion of the Straight Egyptian. The Mongolian Racing horses did not overlap with the Chinese Mongolian horses and were distributed mainly across PC2 (13.2% of the variance).

In summary, although there was some shared ancestry among the Racing populations, this was not widespread among individual animals suggesting that the observed Thoroughbred admixture most likely reflects recent breeding practices. Therefore, it is unlikely to influence detection of long-standing selection signals due to persistent selection over relatively extended time frames. Furthermore, the composite selection signals (CSS) approach used in this study combines component signals to detect only strongly selected regions that have a common signal across the constituent tests[39]. By contrast there is considerable Thoroughbred gene flow in other racehorse breeds such as Quarter Horse, which has a distinct subpopulation bred for racing[5,16]. Consequently, selection signals identified here among the Racing populations (Arabian, Mongolian Racing, and Thoroughbred) were hypothesised to reveal genomic regions contributing to similar genomic architectures that result from convergent evolution towards the racing phenotype and not gene flow.

**Genomic signals of selection among Racing breeds.** To identify genomic regions targeted by selection for the racing phenotype, we compared allele frequency distribution variation among two data sets comprising horses from Racing ($n = 90$) and non-Racing ($n = 483$) breeds (Supplementary Data 2) using a CSS test that combines the XP-EHH, $F_{ST}$ and $\Delta$SAF statistics[39]. Genome-wide distribution of the smoothed CSS score test statistics (-$\log_{10}$ $P$) for comparison of the Racing versus non-Racing populations identified 14 genomic regions with signals of selection, defined as clusters of SNPs (>5) among the top 1% SNPs, on ECA1, ECA2, ECA4, ECA5, ECA7, ECA9, ECA14, ECA17, ECA18, and ECA22

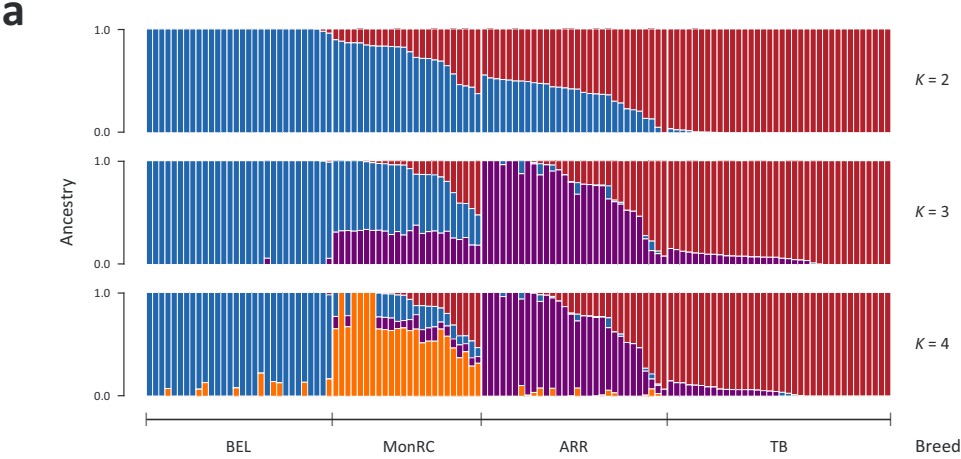

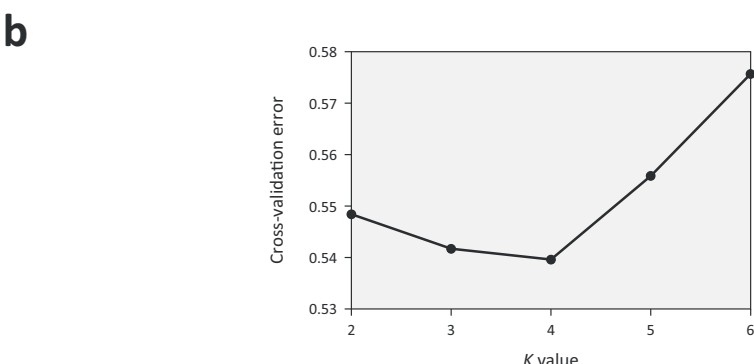

**Fig. 2 Admixture analysis among the Racing breeds using Belgian (BEL) as comparator population. a** bar plots of ancestry composition by admixture for the assumed number of ancestries $K = 2$ to $K = 4$ illustrating the contribution of Thoroughbred (TB) ancestry in the Arabian (ARR) and Mongolian Racing (MonRC) populations. Individual TB contributions to ARR and MonRC are provided in (Supplementary Data 1) **b** Cross-validation error plot for $K = 2$–6 modelled ancestral populations.

(Supplementary Data 3, Fig. 3). Signals on ECA1, ECA7, ECA17 and ECA18 were the highest ranking according to the CSS score. The top ranked region (ECA1, 45.34–46.54 Mb) contained the *PCDH15* and *ZWINT* genes, which supports results obtained for these genes in two different Thoroughbred population samples[15,16]. A selection signature at this genomic region was also previously detected in the same Thoroughbred sample cohort used here with different statistical approaches ($d_i$, H, H$_{12}$, and Tajima's D)[5,7].

There was considerable overlap with selection signals previously reported in a range of athletic horse breeds and the selection signals also overlapped with QTLs identified in GWAS for racing performance traits in Thoroughbreds[43,44] (Table 1). Notably, the second ranked region (ECA7: 40.44–42.86 Mb) containing the *NTM* gene, coincided with the top GWAS peak identified from a comparison of Thoroughbreds that had raced and Thoroughbreds that had never had a racecourse start[44], and a selection signal on ECA2 (ECA2: 100.3–101.78 Mb) overlapped with a GWAS peak for measured speed traits in juvenile Thoroughbreds[43].

**Functional enrichment among genes in selected regions**. To assess enrichment of functional ontologies in selected regions for Racing, we assigned functional annotation to all the genes in the regions defined by the top 1% SNPs (including those with <5 SNPs) using the DAVID functional annotation tool[45] (Table 2,

Supplementary Data 4). A challenge to employing functional enrichment tools to such gene sets is the presence of gene family clusters in the same chromosomal region; for example, the gamma-aminobutyric acid signalling pathway (GO:0007214) and GABAergic synapse (GO:0098982) genes (*GABRA1, GABRA6, GABRB2, GABRG2, SLC12A2*) are, except for *SLC12A2*, located at a single locus on ECA14. Nonetheless, there were several exercise-relevant gene ontology terms enriched among the genes that were located on different chromosomes including heart looping (GO:0001947; *BBS4, BBS5, SETDB2, KIF3A*), cardiac muscle tissue morphogenesis (GO:0055008; *BMP2, MYLK2, XIRP2*), cellular respiration (GO:0045333; *FASTKD1, COX4I2, TBRG4*), skeletal muscle satellite cell differentiation (GO:0014816; *MEGF10, MYLK2*), and glycolysis/gluconeogenesis (GO:0006094; *ADPGK, ALDH7A1, G6PC2, PKM*) (Table 2).

**Genomic signals of selection in Arabian, Thoroughbred, and Mongolian Racing breeds**. We evaluated the overlap between the Racing selection signals and selection signals identified when each of the Racing breeds was analysed separately (Table 1, Supplementary Data 3). The overlap among the Racing selection signals with selection in the Thoroughbred (only) was clear, with nine of the 14 clusters also detected in the Thoroughbred versus other breeds analysis (Supplementary Fig. 4, Supplementary Data 3). There were six selected regions unique to Thoroughbred on ECA1, ECA21, ECA28 and ECA30.

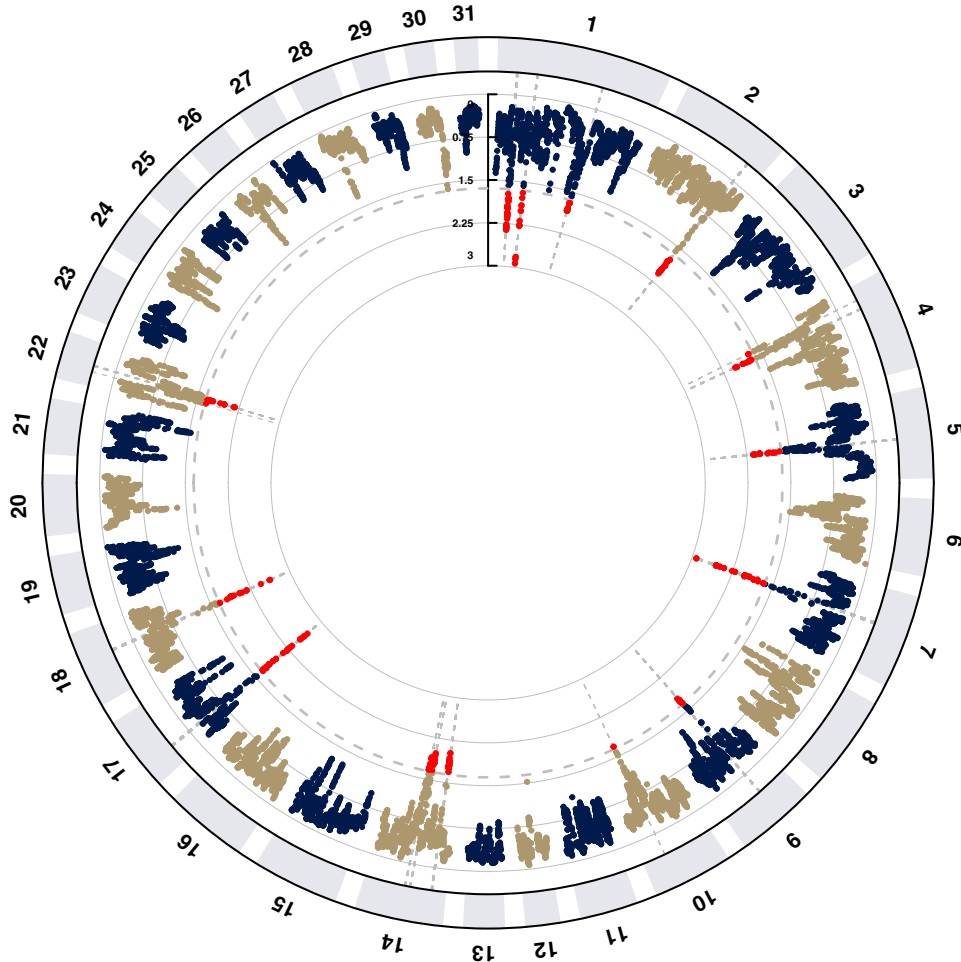

**Fig. 3 Manhattan plot for the results of the composite selection signals (CSS) analysis to detect targets of selection among Racing breeds ($n = 90$ horses) when compared with non-Racing breeds ($n = 483$ horses).** Non-Racing breeds had the following phenotypes: draft, small size, riding, driving, gait. Detailed information for the breeds used is provided in Supplementary Data 2. The results were obtained by averaging the CSS scores of SNPs within 100 kb sliding windows. The dashed grey line indicates the genome-wide (1% SNPs) threshold of the empirical scores and the top SNPs are indicated by red dots.

There was also considerable overlap among the Racing selection signals with selection in the Arabian (only), with seven of the 14 clusters also detected in the Arabian versus other breeds analysis (Supplementary Fig. 5, Supplementary Data 3). There were 11 selected regions unique to Arabian on ECA2, ECA3, ECA8, ECA12, ECA19 and ECA23. The Arabian (only) selected regions contained some recognised equine exercise relevant genes including *COX4I1*[29,46,47], *PPARGC1A*[47,48] and *DMRT3*[49]; all three of these genes have been identified within runs of homozygosity in several horse breeds[50].

Only two Racing selection signals overlapped with Mongolian Racing (only) selection signals, with 15 clusters unique to Mongolian Racing. Three regions unique to Mongolian Racing stood out as having very strong signals of selection (ECA5, 26, 28) (Fig. 4, Supplementary Data 3). The top ranked region spanned 5.6 Mb on ECA5 (43.32–48.93 Mb) and contained an eSNP (rs69550318) for the *SELENBP1* gene that has been identified among the top 10 trans eQTL among genes expressed in Thoroughbred skeletal muscle[51]. In human endurance athletes *SELENBP1* is differentially expressed in blood in response to administration of human recombinant erythropoietin suggesting a potential role in the regulation of haematopoiesis[52,53]. The top ranked SNP for the overall CSS score and XP-EHH statistic was located within the *TBX15* gene that functions in skeletal development of the limb, vertebral column[54], and shoulder and

pelvic girdles[55]. Conformation is a key phenotype on which racehorses are selected and the axis of the pelvis has been shown to be associated with injury risk and performance in Thoroughbreds[56]. *TBX15* also plays a major role in skeletal muscle fibre type differentiation and regulates the metabolism of glycolytic myofibres[57] and white adipocytes[58] especially in the browning of adipocytes and has been considered a target for the treatment of obesity[59,60]. In this regard, we previously proposed that adipocyte browning may be a key contributor to the equine athletic phenotype[22]. Furthermore, *TBX15* is among the top ranked differentially expressed downregulated genes in Thoroughbred skeletal muscle in response to training[24], implicating it as central to adaptation to the exercise stimulus.

The second ranked CSS region contained the top ranked SNP according to the ΔSAF statistic that was 16 kb from the closest gene, *PPARA*, which has a major role in exercise and training and is associated with elite human endurance athlete status[61,62]. The highest-ranking SNPs on ECA26 encompassed three genes (*NRIP1, BTG3,* and *CHODL*) all of which may be candidate genes for exercise adaptation[63–68]. The highest-ranking SNPs according to the $F_{ST}$ statistic were on ECA7 within the *NDUFB7* and *CACNA1A* genes. *NDUFB7* encodes a structural subunit of complex I of the mitochondrial respiratory chain and mutations in the gene have been observed to cause hypertrophic cardiomyopathy and lactic acidosis[69]. In a

**Table 1 Selection signals for Racing.**

| Selection signals identified in Racing breeds | | | | | Overlapping selection signals identified in individual breed | | | |
| --- | --- | --- | --- | --- | --- | --- | --- | --- |
| Chr | Cluster region (Mb) | CSS score | Overlapping selection signals in athletic and racing breeds and breeds related to racing breeds | Overlapping GWAS hits (TB) | ARR | MonRC | TB | Region searched in WGS |
| 1 | 45.34–46.54 | 2.930 | Thoroughbred[5,7,15,16], Swiss Warmblood[5], Racing Quarter Horse[142] | | | | ✓ | ✓ |
| 7 | 40.44–42.86 | 2.923 | Warmblood[143], Akhal Teke, Arabian, Mongolian, Standardbred[5] | Race starts[44] | ✓ | | ✓ | ✓ |
| 17 | 21.03–23.27 | 2.681 | Thoroughbred[5,7,15], Arabian[7], Hanoverian, Mongolian, Quarter Horse[5] | | | ✓ | ✓ | ✓ |
| 18 | 48.08–50.4 | 2.624 | Thoroughbred[15], Racing Arabian[7], Hanoverian[5] | | ✓ | | ✓ | ✓ |
| 1 | 21.68–23.24 | 2.359 | Thoroughbred[7,15], Quarter Horse, Standardbred[5] | | ✓ | | ✓ | ✓ |
| 22 | 23.37–24.4 | 2.175 | Thoroughbred[7], Arabian, Swiss Warmblood[5] | | ✓ | | ✓ | ✓ |
| 5 | 51.44–52.88 | 2.135 | Turkemen[7] | | ✓ | | | ✓ |
| 2 | 100.3–101.78 | 2.076 | Racing Arabian[7] | Speed[43] | ✓ | | | ✓ |
| 4 | 26.75–27.74 | 2.026 | Thoroughbred Iranian Arabian[7], Hanoverian[5] | | | | | |
| 14 | 15.51–17.07 | 2.023 | Thoroughbred[5] | | | | ✓ | |
| 14 | 41.12–42.22 | 1.981 | Thoroughbred[5,7,23], Racing Arabian[7], Mongolian, Quarter Horse, Swiss Warmblood[5] | | | | ✓ | |
| 14 | 46.82–47.91 | 1.898 | Thoroughbred[5,23], Racing Arabian[7] | | | | | |
| 1 | 122.12–122.93 | 1.841 | Thoroughbred[7,15], Straight Egyptian[7] | | ✓ | | ✓ | ✓ |
| 9 | 35.36–36.69 | 1.775 | Racing Quarter Horse[142] | | | | | |

Selection signals in the Racing breeds (n = 90 horses) versus non-Racing breeds (n = 483 horses) composite selection signals (CSS) analysis showing overlaps with targets of selection previously identified in breeds putatively ancestral to Thoroughbred (Turkemen, Akhal-Teke) and other athletic (Swiss Warmblood, Hanoverian, Warmblood) and racehorse (Iranian Arabian, Quarter Horse, Racing Arabian, Racing Quarter Horse, Standardbred, Straight Egyptian) breeds; overlaps with GWAS hits for racecourse starts and measured speed in Thoroughbreds (TB); overlaps with individual Racing breeds CSS results generated in this study (Arabian—ARR, Mongolian Racing—MonRC, Thoroughbred—TB; genomic regions screened in whole genome sequence (WGS) data for putative functional sequence variants in candidate exercise-related genes).

**Table 2 Summary of exercise-relevant gene ontology terms enriched among genes in Racing selected regions.**

| | | DAVID functional annotation | | | | KEGG Pathway |
| --- | --- | --- | --- | --- | --- | --- |
| | | Biological process | | | | |
| Chrom. | Cluster region (Mb) | Heart looping | Cardiac muscle tissue morphogenesis | Cellular respiration | Skeletal muscle satellite cell differentiation | Glycolysis/ gluconeogenesis |
| ECA17 | 21.03–23.27 | SETDB2 | | | | |
| ECA18 | 48.08–50.4 | BBS5 | XIRP2 | FASTKD1 | | G6PC2 |
| ECA22 | 23.37–24.4 | | MYLK2 | COX4I2 | MYLK2 | |
| ECA14 | 41.12–42.22 | KIF3A | | | | |
| ECA14 | 46.82–47.91 | | | | MEGF10 | ALDH7A1 |
| ECA1 | 122.12–122.93 | BBS4 | | | | ADPGK |
| | | | | | | PKM |

The complete set of results is provided in Supplementary Data 4.

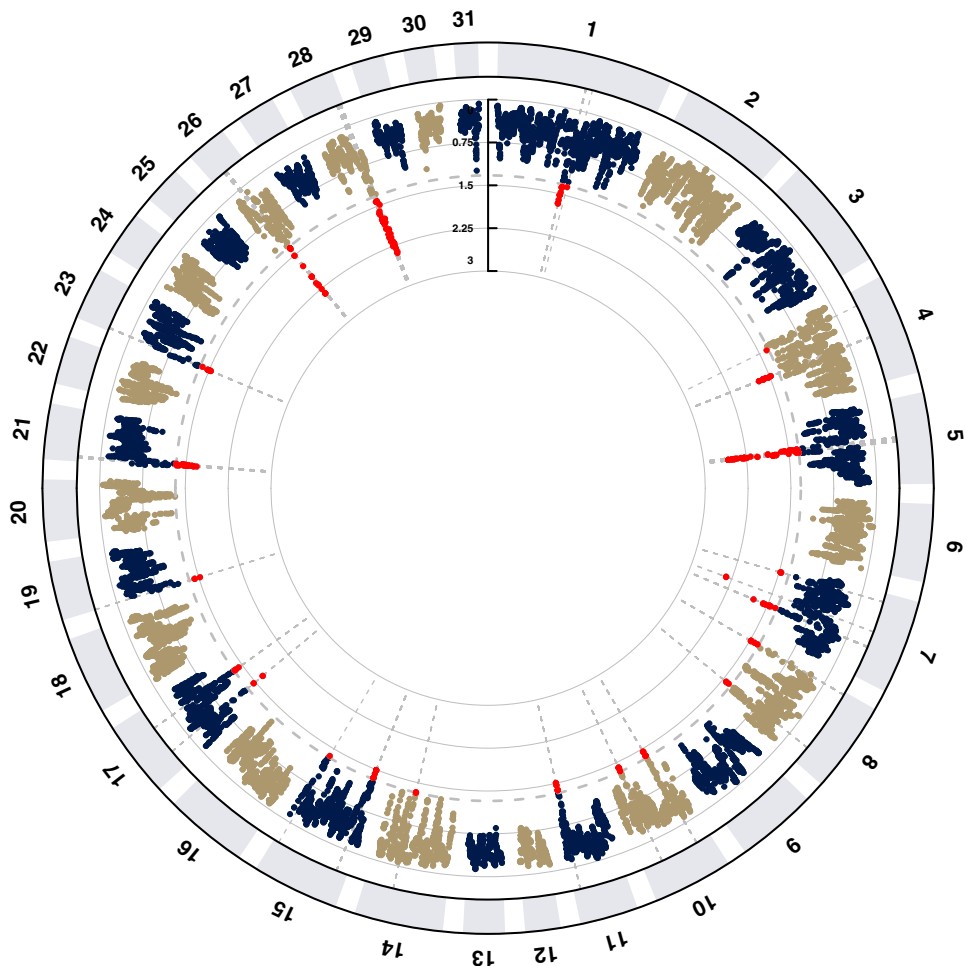

**Fig. 4 Manhattan plot for the results of the composite selection signals (CSS) analysis to detect targets of selection among Mongolian Racing (MonR) horses ($n = 24$ horses) when compared with other breeds ($n = 549$ horses).** The results were obtained by averaging the CSS scores of SNPs within 100 kb sliding windows. The dashed grey line indicates the genome-wide (1% SNPs) threshold of the empirical scores and the top SNPs are indicated by red dots.

cancer model, *NDUFB7* expression is directly modulated by PPARA[70] and mutations in *CACNA1A* cause congenital ataxia in humans[71,72].

**Challenges to identifying candidate genes**. It is difficult to rely solely on selection signals arising from SNP genotyping array data to pinpoint the gene(s) that may have been subject to natural or human-mediated selection. Almost half of the selection signals identified in the four analyses spanned >1 Mb; the largest region was for Mongolian Racing (ECA5: 43.32–48.93 Mb, 5.6 Mb), followed by three regions on ECA17, ECA7, and ECA18 for Thoroughbred (3.5 Mb to 2.9 Mb in size); the largest selection signal for Arabian was 1.8 Mb on ECA3 (37.45–39.25 Mb); and the largest region for the Racing breeds was the second highest ranked selection signal on ECA7 (2.4 Mb). For all analyses, there

was a positive relationship between CSS score and the size of the region identified ($r^2 = 0.6$). Therefore, the most strongly selected regions (and regions of greatest interest) were the largest and in most cases these regions contained sizeable numbers of genes. Proximity of a top ranked SNP (using CSS or any of the individual test statistics) to a gene may be informative to some degree; however, linkage disequilibrium extends across large regions in horse populations[17], the equine SNP genotyping arrays exhibit ascertainment bias[73,74] being designed to assay genetic variation among the main European and North American breeds and, as illustrated above, many genes in each detected region exhibit biological functions easily interpretable as affecting exercise physiology. Therefore, to better characterise genes subject to selection for exercise traits, we used additional methods that leveraged transcriptomics data to prioritise SNPs proximal to DEGs in equine skeletal muscle, and WGS data from a cohort of (mostly) Asian horses.

**Integrative genomics**. Differential patterns of gene expression are the key determinants of phenotype, and integration of transcriptomics and genetic data has been successfully applied to understand the molecular basis of exercise adaptation[75,76]. Here, to refine the SNPs from the population genomics analyses, we integrated these data with DEG sets derived from Thoroughbred skeletal muscle RNA-seq data that distinguish exercise (untrained exercise, UE) and training (trained rest, TR) response transcriptomes[24]. For computational efficiency, the gene lists were refined to include DEGs with $P_{adj.} < 10^{-12}$ (UE) and $P_{adj.} < 10^{-4}$ (TR), which resulted in 407 (UE) (Supplementary Data 5) and 230 (TR) (Supplementary Data 6) DEGs .

The R software package gwinteR[77] was used to determine whether genomic regions containing SNPs that are proximal to genes within the DEG sets were enriched for significance in the CSS analysis for Racing versus non-Racing breeds. The numbers of statistically significant SNPs pre- and post-data integration are summarised in Supplementary Data 7. In terms of SNP enrichment ($P_{perm.} < 0.1$), the integrative analysis was effective for the two input DEG sets. Using a search window that iteratively increased in size from 10–100 kb up and downstream of the genes of interest, a search space of ±100 kb produced the highest number of significantly enriched SNPs with the lowest probability of being significant by chance when compared to a null distribution of 1,000 sets of SNPs randomly sampled from the CSS dataset. SNPs within ±100 kb of a DEG were therefore selected as the target SNP sets to generate new $q$-values. Gene loci associated with enriched CSS SNPs are provided in Supplementary Data 8. Two genes (*LPIN1, LRRC3B*) were enriched for SNPs ($q < 0.05$) in the exercise response (UE) gene set and three genes (*CBR4, SYNDIG1, MYOM2*) were enriched for SNPs ($q < 0.05$) in the training response (TR) gene set. Five genes (*HMOX1, KTN1, MYLK2, NEO1, TUBA4A*) were common to both outputs ($q < 0.1$) and two of these (*NEO1, MYLK2*) were located within the selected regions defined by the top 1% CSS SNPs.

Since there was less overlap between the Mongolian Racing selection signals and the Racing selection signals than there was for the Thoroughbred and the Arabian, we separately integrated the Mongolian Racing CSS SNPs in the context of the skeletal muscle DEGs to refine the gene sets (Supplementary Data 9). Again, using 100 kb windows, three genes (*PPP2R3A, PELO, GLB1*) were enriched for SNPs ($q < 0.05$) in the exercise response (UE) gene set and three genes (*TBX15, KHDRBS3, VEGFA*) were enriched for SNPs ($q < 0.05$) in the training response (TR) gene set (Supplementary Data 10). In addition, three genes (*MAP7D1, STAC3, VEGFA*) were common among the two outputs ($q < 0.1$); however, none of these was located within the selected regions

defined by the top 1% CSS SNPs. Four DEGs with localised SNP enrichment were located in the top-ranked CSS region for the UE and TR gene sets (*APH1A, ATP1A1*, UE; *CA14, TBX15*, TR) and four were among the other CSS regions (*ANKRD23, IFI30, RAB30*, UE; *NXN*, TR). The five most significant SNPs for the TR gene set were upstream and within the *TBX15* gene. The most significant SNP for the UE gene set was in *PPP2R3A*.

**Whole genome resequencing and variant calling**. To identify gene variants with putative functional effects that may be the targets of selection we generated WGS data for 70 horse samples (Supplementary Fig. 6, Supplementary Data 11). A total of ~652 billion 150 bp paired-end reads were generated, with an average depth of 30.13× per individual animal and an average genome coverage of 99.58% (Supplementary Data 12). We obtained 3,846,455 and 3,511,329 polymorphic variants on average per sample after mapping with SAMtools and GATK, respectively, of which 3,177,005 were identified using both methods (Supplementary Data 13). After combining all SNPs from 70 animals, a final set of 24.41 million unique SNPs was retained (3.18 million/individual animal), along with 2.03 million insertion/deletion polymorphisms (indels). Among the ~2 million SNPs on the MNEc2M equine high-density SNP genotyping array[74], on average 315,491 SNPs were identified in the sequenced samples with an average of 99.83% genotyping concordance, which demonstrates the reliability of our SNP calling (Supplementary Data 14).

**Identification of sequence polymorphisms in exercise relevant genes**. To generate a panel of sequence polymorphisms to test for alleles with significant deviations in frequency between different subgroups of horses, we focused on identifying protein-coding variants in candidate genes within the selection signal regions and genes identified from the integrative analysis. We focused on polymorphisms (SNPs and small indels) with moderate minor allele frequencies (MAF >0.1) as we did not expect this approach to identify rare small-effect variants. In addition, we did not expect to identify severely deleterious mutations, and therefore the search was not limited to variants with a predicted high effect on modifying gene function.

For the Racing breeds, the eight highest ranked selected regions and 11 significant regions from the integrative analysis (five common to UE and TR, including two that also overlapped with CSS; three unique to UE; three unique to TR) were used to search for putative functional variants with the WGS data (Table 3). Among the searched regions, for validation we chose high-effect variants in four candidate genes and moderate effect variants in 14 candidate genes (Supplementary Data 15). Three regions did not contain any variants that met the prioritisation criteria. Notably absent were variants in the top ranked CSS region on ECA1 that contained *PCDH15* and *ZWINT*. *PCDH15* has been associated with lipid phenotypes[78], but is best known for association with deafness[79] and is not a compelling candidate gene. On the other hand, the ZW10 interactor protein, encoded by *ZWINT*, functions in neurotransmitter release and in rodents mediates negative behaviour induced by neuropathic pain[80], which may be relevant to exercise[81]. We previously reported a sequence tag <5 kb from *ZWINT* among the most differentially expressed downregulated transcripts in the training-response skeletal muscle transcriptome in the horse[25] implicating the locus as functionally relevant to exercise. The absence of identified gene-specific variants in this region may be explained by the focus here on the identification of common protein-coding variants, which precludes the identification of sequence variants in genomic regulatory elements, copy number variants, and

**Table 3 Genomic regions chosen to search for high and moderate effect variants in the whole genome sequence data.**

| Chr. | Region start | Region end | Region size (Mb) | Analysis method | No. SNPs high effect MAF ≥ 0.1 | No. SNPs mod effect MAF ≥ 0.1 | Candidate gene(s) with high/mod effect variant | Cohort |
|---|---|---|---|---|---|---|---|---|
| 1 | 21680000 | 23240000 | 1.56 | CSS | 2 | 0 | – | Racing |
| 1 | 45340000 | 46540000 | 1.20 | CSS | 0 | 2 | – | Racing |
| 1 | 122120000 | 122930000 | 0.81 | TR, UE, CSS | 1 | 3 | PKM | Racing |
| 2 | 66441005 | 66747911 | 0.31 | TR | 0 | 2 | CBR4 | Racing |
| 2 | 100300000 | 101780000 | 1.48 | CSS | 0 | 2 | INTU | Racing |
| 4 | 50367171 | 51430398 | 1.06 | UE | 0 | 4 | HDAC9 | Racing |
| 5 | 51440000 | 52880000 | 1.44 | CSS | 1 | 4 | SLC16A1 | Racing |
| 6 | 8463018 | 8667520 | 0.20 | TR, UE | 0 | 0 | TUBA4A | Racing |
| 7 | 40440000 | 42860000 | 2.42 | CSS | 0 | 1 | NTM | Racing |
| 15 | 83051411 | 83375153 | 0.32 | UE | 0 | 4 | LPIN1 | Racing |
| 16 | 58089557 | 58374430 | 0.28 | UE | 0 | 0 | – | Racing |
| 17 | 21030000 | 23270000 | 2.24 | CSS | 1 | 0 | KPNA3 | Racing |
| 18 | 48080000 | 50400000 | 2.32 | CSS | 0 | 5 | G6PC2, FASTKD1, PPIG | Racing |
| 22 | 1076610 | 1464865 | 0.39 | TR | 0 | 2 | SYNDIG1 | Racing |
| 22 | 23370000 | 24400000 | 1.03 | TR, UE, CSS | 0 | 4 | MYLK2 | Racing |
| 24 | 3454073 | 3755465 | 0.30 | TR, UE | 0 | 2 | KTN1 | Racing |
| 27 | 38413050 | 38745592 | 0.33 | TR | 0 | 1 | MYOM2 | Racing |
| 28 | 33915186 | 34121336 | 0.21 | TR, UE | 2 | 1 | HMOX1 | Racing |
| 4 | 55163167 | 56163167 | 1.00 | CSS | 0 | 2 | NPY | MonRC |
| 5 | 46453083 | 49712027 | 3.26 | TR, UE, CSS | 0 | 43 | TBX15, ATP1A1 | MonRC |
| 7 | 64386638 | 65386638 | 1.00 | CSS | 1 | 9 | PRCP | MonRC |
| 9 | 77322133 | 78512097 | 1.19 | TR | 0 | 0 | – | MonRC |
| 15 | 11535656 | 12541448 | 1.01 | UE, CSS | 1 | 25 | ANKRD23 | MonRC |
| 16 | 52279295 | 53369884 | 1.09 | UE | 1 | 20 | GLB1 | MonRC |
| 16 | 72225972 | 73379121 | 1.15 | UE | 0 | 3 | PPP2R3A | MonRC |
| 20 | 43088186 | 44106716 | 1.02 | TR | 1 | 24 | VEGFA | MonRC |
| 21 | 3270978 | 4270978 | 1.00 | CSS | 5 | 59 | COMP | MonRC |
| 21 | 19256258 | 20260886 | 1.00 | UE | 0 | 0 | – | MonRC |
| 26 | 16203156 | 17203156 | 1.00 | CSS | 1 | 12 | CXADR | MonRC |
| 28 | 39787330 | 43768541 | 3.98 | CSS | 6 | 136 | SULT4A1 | MonRC |

Analysis method—rationale for inclusion among search regions from results of either Racing or Mongolian Racing (MonRC) analyses; number of high and moderate (mod) effect variants within the region; candidate exercise-relevant genes containing high or moderate effect variants.

chromatin state modifications that also contribute to the gene regulatory networks underlying complex traits[82,83].

A similar approach was taken to identify putative functional variants within regions identified for the Mongolian Racing analyses. For Mongolian Racing the seven highest ranked selected regions and seven significant regions from the integrative analysis (one common to UE and TR that also overlapped with CSS; four unique to UE including one that overlapped with CSS; two unique to TR) were prioritised (Table 3). For validation, we chose high effect variants in four candidate genes and moderate effect variants in eight candidate genes (Supplementary Data 15).

In total, 32 polymorphisms in 27 genes were selected for validation genotyping on the basis that the variants disrupt the sequence of proteins with central roles in exercise physiology—including key functions associated with muscle, heart, angiogenesis/blood, limb development, metabolism, and neurological tissues. The known biological functions of the genes are summarised in Supplementary Note 1. Of the 32 polymorphisms, 23 SNPs met the assay design criteria and passed post-genotyping quality control and were used in tests of genetic association. Genotypes were generated for independent validation sample sets that were not used for the selection signals analyses.

**Genetic association with the racing phenotype.** We hypothesised that genetic variants targeted by selection for the racing phenotype segregate among horse breeds to influence underlying endophenotypic variation. Genotypes for the panel of 23 SNPs were generated for $n = 267$ horses from six breeds (Arabian, French Trotter, Mongolian Racing, Quarter Horse, Standardbred,

and Thoroughbred) chosen to represent racing breeds, and $n = 249$ horses from eleven breeds (putatively ancestral to Thoroughbred—Akhal Teke, Egyptian Arabian, Moroccan Barb; Chinese Mongolian landrace—Baerhu, Baicha Iron Hoof, Keerqin, Wushen, Wuzhumuqin; sport horses—Connemara, Irish Draught, Dutch Warmblood) representing non-racing breeds (Supplementary Data 16). Additional detail for the breeds is provided in Supplementary Note 2.

In tests of genetic association, SNPs in nine genes were significantly (Bonferroni-adjusted $P < 3.57 \times 10^{-3}$) associated with the racing phenotype (Table 4). Eight were missense variants predicted to have a moderate effect on the protein and one (SLC16A1) that introduces a stop codon was predicted to have a high effect on the protein. We did not expect to identify loss of function mutations, since we expected here to detect alleles that are advantageous for exercise. The introduction of a stop codon may not always disrupt the function of a protein if there is limited truncation or if there is stop codon read through[84].

**Biological functions relevant to exercise among genes significantly associated with racing.** The functional relevance of this gene set is supported by the integrative analyses in which three of the genes (KTN1, MYLK2, and SYNDIG1) were enriched for SNPs among DEGs in the skeletal muscle exercise and training response. A literature search and review of associated gene ontology functions, indicated that this set of genes have roles in muscle (HDAC9, MYLK2), metabolism (FASKD1, G6PC2, GLB1, SLC16A1) and neurobiological (KTN1, NTM, SYNDIG1) functions that are linked to exercise-relevant phenotypes.

**Table 4 SNPs significantly associated with the racing phenotype among global breeds.**

| Chrom. | Position | Allele 1 | F_A | F_U | Allele 2 | CHISQ | Unadjusted P value | Odds Ratios | Gene |
|---|---|---|---|---|---|---|---|---|---|
| ECA7 | 41381993 | A | 0.2715 | 0.4980 | G | 55.94 | $7.49 \times 10^{-14}$ | 0.376 | NTM |
| ECA18 | 48638568 | A | 0.1873 | 0.3956 | G | 54.56 | $1.51 \times 10^{-13}$ | 0.352 | G6PC2 |
| ECA22 | 1289141 | T | 0.0414 | 0.1714 | C | 46.53 | $9.02 \times 10^{-12}$ | 0.209 | SYNDIG1 |
| ECA18 | 49159216 | T | 0.4678 | 0.6741 | C | 44.25 | $2.89 \times 10^{-11}$ | 0.425 | FASTKD1 |
| ECA18 | 49230205 | A | 0.5418 | 0.7117 | G | 31.40 | $2.10 \times 10^{-8}$ | 0.479 | PPIG |
| ECA4 | 51109357 | G | 0.6616 | 0.8117 | A | 28.84 | $7.86 \times 10^{-8}$ | 0.454 | HDAC9 |
| ECA22 | 23460297 | A | 0.3783 | 0.2541 | T | 17.90 | $2.33 \times 10^{-5}$ | 1.786 | MYLK2 |
| ECA5 | 52312504 | G | 0.1184 | 0.2146 | C | 17.21 | $3.34 \times 10^{-5}$ | 0.492 | SLC16A1 |
| ECA24 | 3586665 | G | 0.1816 | 0.2831 | A | 14.96 | $1.10 \times 10^{-4}$ | 0.562 | KTN1 |
| ECA15 | 83175651 | C | 0.0861 | 0.1586 | A | 12.72 | $3.62 \times 10^{-4}$ | 0.500 | |
| ECA2 | 101601341 | A | 0.3783 | 0.4859 | G | 12.19 | $4.82 \times 10^{-4}$ | 0.644 | |
| ECA28 | 34013061 | G | 0.3132 | 0.2329 | C | 8.31 | $3.94 \times 10^{-3}$ | 1.502 | |
| ECA2 | 66547160 | G | 0.1351 | 0.2033 | A | 8.31 | $3.96 \times 10^{-3}$ | 0.612 | |
| ECA27 | 38633488 | T | 0.0947 | 0.1209 | C | 1.82 | 0.178 | 0.761 | |

SNPs that were significant following Bonferroni correction for multiple testing are annotated with the associated gene name. F_A = frequency of allele 1 in Racing breeds, F_U = frequency of allele 1 in non-Racing breeds. Racing breeds horses $n = 267$; non-Racing breeds horses $n = 249$. Racing breeds—Arabian, French Trotter, Mongolian Racing, Quarter Horse, Standardbred, Thoroughbred; non-Racing Breeds—Baerhu, Baicha Iron Hoof, Keerqin, Wushen, Wuzhumuqin, Akhal Teke, Egyptian Arabian, Moroccan Barb, Connemara, Irish Draught, Dutch Warmblood. RS codes for SNPs are provided with additional details in Supplementary Data 15.

Skeletal muscle is a highly plastic tissue that responds to exercise and training stimuli by increasing muscle mass and changing fibre type composition with concomitant mitochondrial functional adaptations[85]. Here, we identified two genes relevant to muscle function that were significantly associated with the racing phenotype, and we consider to be core genes. The HDAC9 gene encodes a protein that inhibits skeletal myogenesis and is involved in heart development[86–89]. In Thoroughbred skeletal muscle HDAC9 is among the top five most significant DEGs downregulated in the exercise response ($\log_2 FC = -2.67$, $P = 1.21 \times 10^{-20}$)[24]. In humans, HDAC9 gene variants are associated with the maximal oxygen uptake ($VO_{2max}$) response to training[90]. Among the racing breeds, the Thoroughbred had the highest frequency (0.65) of the A-allele, which was more than twice the frequency of the allele among the other racing breeds (mean = 0.31) and 3.4× that among the sport horse breeds (0.19). Allele frequencies for all SNPs in each breed are shown in Supplementary Data 17.

MYLK2 encodes a myosin light chain kinase (MYL2) expressed in skeletal muscle. The enzyme has a critical role in muscle contraction, and functions in neuromuscular synaptic transmission, skeletal muscle satellite cell differentiation, regulation of muscle filament sliding and skeletal muscle cell differentiation[91,92]. MYLK2 was the most significantly downregulated gene ($\log_2 FC = -1.31$, $P = 1.37 \times 10^{-22}$) among the 3,241 DEGs in Thoroughbred skeletal muscle following exercise and ranked 6th following a period of training ($\log_2 FC = -1.04$, $P = 1.13 \times 10^{-6}$)[24], higher than MSTN (14th, $\log_2 FC = -2.56$, $P = 1.43 \times 10^{-6}$), a gene with a well-established functional role in exercise[19,29,31,33]. In humans, genetic variants in MYLK are associated with phenotypic responses to exercise-induced muscle damage[93]. Here, the A-allele that was significantly different between racing (0.38) and non-racing breeds (0.25), had, among the racing breeds, the highest frequency in Arabian (0.63) and French Trotter (0.45) and the lowest frequency in Quarter Horse (0.28) and Standardbred (0.28) (Supplementary Data 17). Based on the considerable functional evidence, we propose that genetic variation in HDAC9 and MYLK2 has a critical role in determining the muscle phenotype of racehorses.

The metabolic properties of skeletal muscle are largely influenced by the proportion of slow (type I) and fast (type II) muscle fibres, which are defined by the myosin heavy chain isoforms and characterised by the different densities and functional properties of mitochondria[94]. Within the different fibre types, the glycolytic and oxidative pathways are tightly regulated to ensure an adequate supply of ATP to meet energy demands. The G6PC2 gene encodes a major component of glycolysis[95–97]. Protein-coding variants in the gene are associated with fasting glucose levels in humans and there is strong evidence that G6PC2 is an effector gene for glucose regulation[97]. Among breeds, the G-allele occurred at the highest frequency in Thoroughbred (0.95) and was lowest in Connemara (0.50). The glycolytic requirements for high intensity exercise are likely responsible for the observed variation in this gene among the breeds. The FASTKD1 and PPIG genes are also located in the region exhibiting the selection signal for G6PC2. The FAST kinase domains 1 protein, encoded by FASTKD1, supports mitochondrial homeostasis, and has a critical protective role against oxidant-induced cell death[98–101]. However, the strongest association in racing breeds was with the G6PC2 SNP and its well-established biological function in the regulation of glucose suggests that it could underpin the selection signal at this locus.

One of the best characterised genes for racing performance in Arabian horses is the SLC16A1 gene encoding the solute carrier family 16 member 1 protein that catalyses the movement of lactate and pyruvate across the plasma membrane[8,9,102]. In humans, genetic variants in the gene are used to predict athletic performance, in particular high-intensity exercise, and power ability[103,104]. Here, we have identified a novel variant that is predicted to have a major effect on the resulting protein through the introduction of a stop codon. The value of this variant in prediction of racing performance among Arabian horses requires testing in horses phenotyped for economically relevant racing traits.

Neurobiological functions have regularly featured in equine exercise transcriptomics and genomics research[24,43,105]. Here we identified SNPs in three genes with functions in neurobiology, KTN1, NTM and SYNDIG1. Of particular note is NTM encoding neurotrimin, which functions in brain development, regulates neural growth and synapse formation, and influences learning and memory[106–111]. A GWAS in Thoroughbreds previously identified this locus as the most significantly associated with the number of racecourse starts[44]. NTM also ranks among the top 10 genes positively selected during horse domestication[112] suggesting that equine neurological systems associated with

domestication may overlap with adaptive traits that are required for racing. Here, the *NTM* SNP was the most significantly associated ($P = 7.49 \times 10^{-14}$) with the racing breeds and among all breeds the highest frequency of the racing allele was in the Thoroughbred (0.89).

In humans, *KTN1* gene variants are strongly associated with *KTN1* gene expression in the putamen and the volume of the putamen[113], a region of the forebrain belonging to the basal ganglion that influences motor behaviours including motor planning and execution, motor preparation, amplitudes of movement and sequences of movement[113–120]. Here, the *KTN1* G-allele was <1% in Thoroughbreds but had an average frequency of 0.22 in the other racing breeds, was 0.34 in the ancestral breeds and 0.21 in the sport horse breeds. Selection for the racing allele in breeds other than the Thoroughbred may be valuable in improving locomotor functions critical to racing. For *SYNDIG1*, the product of which regulates the development of excitatory synapses[121–123], we observed that selection may already have fixed the exercise-favoured variant in racing breeds; the T-allele was absent in Thoroughbred, Arabian and Akhal Teke and was observed at a low frequency in the other breeds, with the highest occurrence in Connemara (0.29) and Irish Draught (0.25).

**Genes associated with racing in Mongolian horses**. Considering the difference in the selection signals profile of the Mongolian Racing horses (compared to the results from the Racing breeds), we also performed tests of genetic association for eight SNPs in a cohort of Mongolian horses selected by herdsmen for racing by comparing the genotypes to a set of Chinese Mongolian horses that are not used for racing (Supplementary Data 16). The *GLB1* SNP was significantly (Bonferroni-adjusted $P < 0.006$) associated with the racing phenotype among Mongolian horses (Table 5). The protein encoded by *GLB1*, beta-galactosidase, has a role in several metabolic pathways and is the most widely used bio-marker for senescent and aging cells[124]. There are a number of *GLB1* related disorders[125] including a disruption of normal skeletal morphologies[126] and cardiomyopathies.

**Genes associated with racing performance in Thoroughbred horses**. To test for genetic association with racing traits among Thoroughbreds, we partitioned a large archive of samples ($n = 1134$) into three groups: horses classified as elite, horses that had raced but had never won a race, and horses that were unraced (Supplementary Data 18). Among a cohort of horses that had raced in North America, the *MYLK2* SNP was significantly ($P < 0.005$) associated with elite racing performance, but it was not associated with the trait among Australian ($P = 0.43$) or European ($P = 0.47$) horses (Supplementary Data 19). We have previously observed regional-specific variation for racing performance among Thoroughbreds[23], which may be due to different

selection pressures for the various dynamics in each racing ecosystem. Among European Thoroughbreds, the *NTM* SNP was suggestive of association with the occurrence of a racecourse start ($P = 0.01$), and although it did not meet the threshold for significance following correction for multiple testing, the occurrence of this locus in a previous GWAS[44], and the observation of the highest frequency of the SNP among Thoroughbreds, strongly implicates *NTM* as an economically important gene in the Thoroughbred.

## Conclusions

Adaptation driven by strong directional selection at key loci is likely to be particularly important in livestock species that are subject to management-based selection decisions[37,127]. Here, we provide evidence for major-effect variants in shaping the racing phenotype in horse populations. We have demonstrated that allelic variants in a core set of fundamental exercise-relevant genes segregating among horse populations likely underpin an array of adaptations required for racing. Genes including *G6PC2*, *HDAC9*, *KTN1*, *MYLK2*, *NTM*, *SLC16A1* and *SYNDIG1*, with roles in muscle, metabolism, and neurobiological functions, appear to be central to shaping the racing phenotype in horses. These results are likely to inform genome-enabled improvement of horse racing populations and highlight genes of interest for athletic traits in other species for which exercise adaptation is desired.

## Methods

**Ethics statement**. Samples genotyped in this study were collected with informed owner's consent for commercial genetic testing and approved for use in research. As such, institutional animal research ethics was not required. Approval for collection and the movement of genetic material for the whole genome sequencing analyses was granted by Inner Mongolia Agricultural University Animal Research Ethics Committee and Mongolian University of Science and Technology.

**Selection signals analysis samples**. Mongolian Racing: Genotypes were generated for $n = 24$ Mongolian Racing horse samples using the GGP Equine SNP70 genotyping array and were used as the primary Mongolian Racing cohort (Supplementary Data 2). Samples were collected in Khentii province, Mongolia, in 2018. No pedigree/identification or racing performance data was available for the horses. Horses were selected for sampling based on the herdsmen's knowledge of relatedness, from herds of horses bred and owned by the Ajnai Sharga Horse Racing team. Mongolian populations that had been previously genotyped[4] were used for the breed-specific PCA and included horses from the Abaga Black, Baicha Iron Hoof, Sanhe, Wushen and Wuzhumuqin populations.

Arabian: Genotypes were generated for $n = 30$ Arabian horse samples using the GGP Equine SNP70 genotyping array and were used for the selection signals analysis only. Samples were collected in United Arab Emirates in 2014. All horses were bred for racing competitions. No additional pedigree/identification or racing performance data was available for the individual horses. To confirm the racing phenotype for this sample, a PCA was performed using this sample and an additional $n = 152$ Arabian horses that had been previously genotyped[7] and had recorded performance uses (racing, endurance, show). The sample of horses genotyped in this study was confirmed to be distributed in the PCA space shared predominantly with racing Arabians (Supplementary Fig. 7). For the breed-specific

**Table 5 SNPs significantly associated with the racing phenotype among Mongolian horses.**

| Chrom. | Position | Allele 1 | F_A | F_U | Allele 2 | CHISQ | Unadjusted *P* value | Odds ratios | Gene |
|---|---|---|---|---|---|---|---|---|---|
| ECA16 | 52850203 | G | 0.6556 | 0.4625 | A | 9.77 | $2.00 \times 10^{-3}$ | 2.212 | *GLB1* |
| ECA15 | 12036913 | C | 0.1413 | 0.2563 | T | 5.04 | $2.50 \times 10^{-3}$ | 0.478 | |
| ECA16 | 72765824 | G | 0.1957 | 0.1281 | A | 2.43 | 0.119 | 1.656 | |
| ECA4 | 55899514 | G | 0.6667 | 0.4213 | C | 1.44 | 0.231 | 2.747 | |
| ECA7 | 64406448 | A | 0.1304 | 0.1125 | G | 0.21 | 0.650 | 1.183 | |
| ECA26 | 16940454 | A | 0.1000 | 0.0917 | G | 0.05 | 0.817 | 1.101 | |
| ECA21 | 3979432 | T | 0.0978 | 0.0924 | C | 0.02 | 0.880 | 1.065 | |
| ECA5 | 49201314 | A | 0.4222 | 0.4174 | C | 0.01 | 0.936 | 1.020 | |

The SNP that was significant following Bonferroni correction for multiple testing is annotated with the associated gene name. F_A = frequency of allele 1 in Mongolian Racing, F_U = frequency of allele 1 in Chinese Mongolian breeds. Mongolian Racing horses $n = 46$; non-Racing Chinese Mongolian breeds horses $n = 121$. RS codes for SNPs are provided with additional details in Supplementary Data 15.

PCA (Supplementary Fig. 2) Arabian horses from different geographic regions were used. Although there are numerous different subtypes of Arabian horses based on geographic origin and performance use, for simplicity, the racing Arabian horses used in this study are referred to as Arabian. Additional detail for the breed is provided in Supplementary Note 2.

Thoroughbred: The Thoroughbred cohort ($n = 36$) had been previously genotyped[5] and included horses originating in USA and Europe (Supplementary Data 2).

Racing breeds: For the CSS analysis the Mongolian Racing, Arabian and Thoroughbred populations described above were used as the Racing cohort.

Non-Racing breeds: Horses representing 21 diverse horse breeds that have not been bred for racing were used as the comparator cohort to represent non-Racing breeds for the CSS analyses. Samples had been previously genotyped[5] and included Akhal Teke ($n = 20$), Belgian ($n = 30$), Clydesdale ($n = 24$), Caspian Pony ($n = 16$), Exmoor ($n = 24$), Fell Pony ($n = 21$), Franches-Montagnes ($n = 19$), Hanoverian ($n = 15$), Miniature ($n = 25$), Mangalarga Paulista ($n = 14$), Morgan ($n = 43$), New Forest Pony ($n = 15$), Norwegian Fjord ($n = 21$), North Swedish Horse ($n = 19$), Percheron ($n = 42$), Peruvian Paso ($n = 21$), Paint ($n = 25$), Saddlebred ($n = 25$), Shetland ($n = 27$), Shire ($n = 23$), Tuva ($n = 15$) (Supplementary Data 2).

**Whole-genome resequencing samples.** Horses from breeds indigenous to Mongolia and China, and horses from breeds imported to China were used for generation of whole-genome sequence (WGS) data. Samples from indigenous horses ($n = 50$) were obtained from rural localities across China and Mongolia (Supplementary Fig. 6). Additional samples ($n = 20$) were collected from Akhal-Teke ($n = 5$), Arabian ($n = 5$), Shetland Pony ($n = 5$), Friesian ($n = 2$), Clydesdale ($n = 2$) and Russian Draft ($n = 1$) (Supplementary Data 11). No pedigree information was available for these horses, but sampling of related animals was avoided based on the information provided by horse owners and local herdsmen.

**Validation of association with racing samples.** A set of breeds was genotyped for the SNP panel for tests of association with racing and to determine the prevalence of the racing alleles in racing (trotting – French Trotter $n = 44$, Standardbred Trotter $n = 41$; sprinting – Quarter horse, $n = 23$; endurance – Arabian, $n = 63$ and endurance/sprint – Thoroughbred, $n = 50$), sport horse (Dutch Warmblood $n = 42$, Irish Draught $n = 28$, Connemara Pony $n = 18$) and putatively ancestral to Thoroughbred (Akhal Teke $n = 12$, Moroccan Barb $n = 17$, Egyptian Arabian $n = 11$) populations (Supplementary Data 16).

A set of $n = 46$ Mongolian horses, sampled in Khentii Province, that were among a herd of horses bred for racing and five Chinese Mongolian breeds not known to be selected for racing (Baerhu $n = 15$, Baicha Iron Hoof $n = 21$, Keerqin $n = 22$, Wushen $n = 15$, Wuzhumuqin $n = 48$) were used for validation of the SNP panel for Mongolian Racing.

A set of $n = 1134$ Thoroughbreds partitioned into animals that had (1) won an elite (Group or Listed) race, (2) raced but never won, and (3) never raced, were used for validation of the SNP panel among Thoroughbred racehorses (Supplementary Data 18). The horses were registered Thoroughbreds born in Great Britain and Ireland, Australia, and the USA.

**Genotype QC and datasets.** In total, SNP array-derived genotypes for $n = 574$ horses from 24 distinct populations (Supplementary Data 2) were used, including $n = 30$ Arabian and $n = 24$ Mongolian Racing horses that were genotyped using the GGP Equine SNP70 genotyping array in this study. The $n = 30$ Arabian horses were selected from a set of $n = 70$ horses based on limited relationship ($\hat{\pi} < 0.25$). Genotype data (50,042 autosomal SNPs generated using the Illumina EquineSNP50 genotyping array) for $n = 520$ horses were accessed from www.animalgenome.org/repository/pub/UMN2013.0125[5] including Thoroughbred and horses representing populations that were not specifically bred for racing (non-Racing). These data were merged with genotypes generated in this study and SNP QC and filtering were performed across populations on the merged data set. Individual SNPs with >10% missing data and MAF < 0.01 were removed, resulting in 36,767 SNPs for analyses.

Genotypes for $n = 30$ Arabian horses genotyped in this study were combined with publicly available genotypes for $n = 378$ Arabian horses obtained from the Mendeley Data resource (https://doi.org/10.17632/mkk5khxrbp.3)[7]. After removal of individual SNPs with >10% missing data and MAF < 0.01, 35,292 SNPs remained for PCA analysis.

Genotypes for $n = 24$ Mongolian Racing horses genotyped in this study were merged with publicly available genotypes for $n = 100$ Chinese Mongolian horses obtained from the Open Science Framework (https://osf.io/2xvqf/quickfiles)[4]. After removal of individual SNPs with >10% missing data and MAF < 0.01, 60,994 SNPs remaining for PCA analysis.

**Principal component analysis.** To visualise genetic relatedness among the populations, principal component analysis (PCA) was performed using smartPCA from the EIGENSOFT package (version 4.2)[128]. One outlier (Shire, ID:SH144) was identified and excluded from further analyses. The outlier was separated from the breed cluster and had also been identified as an outlier in the study from which the data were obtained[5].

**Admixture analysis among Arabian and Mongolian Racing populations.** For the analysis of population substructure, model-based clustering was performed using the software package ADMIXTURE[129]. The model assigns ancestry based on a predefined number of $K$ ancestral populations. Individuals are assigned to $K$ clusters based on allele frequencies and the proportion of ancestry from each population is estimated. The analysis was performed for $K$ ranging from 2–6. Unsupervised modelling was used to predict allele frequencies in four ancestral genetic lineages ($K = 4$) and each animal's genome was partitioned and proportionally assigned to one of the four lineages. The outputs from the analysis were visualized using pophelper 2.3.1[130].

**Identification of selection signals.** Composite selection signals (CSS) analyses[39,131,132] were performed for four different comparison sets: (1) Racing ($n = 90$; $n = 30$ Arabian, $n = 36$ Thoroughbred, $n = 24$ Mongolian Racing) *versus* non-Racing breeds ($n = 483$); (2) Arabian ($n = 30$) *versus* all other breeds ($n = 544$); (3) Thoroughbred ($n = 36$) *versus* all other breeds ($n = 537$); (4) Mongolian Racing ($n = 24$) *versus* all other breeds ($n = 549$). Details for the comparator cohorts are provided in Supplementary Data 2.

The CSS approach is among several composite approaches to successfully identify genes under selection for monogenic and polygenic traits in livestock[39,133]. CSS uses fractional ranks of constituent tests allowing a combination of the evidence of historical selection from different population genetic tests of selection. We used the fixation index ($F_{ST}$), the change in selected allele frequency ($\Delta SAF$) and the cross-population extended haplotype homozygosity (XP-EHH) tests combining each test statistic into one composite CSS per SNP. $F_{ST}$ statistics were computed as the differentiation index between the target population/s of interest (as selected) and the contrasting/reference population/s (as non-selected), and the XP-EHH and $\Delta SAF$ statistics were computed for the selected population/s against the reference population (as non-selected).

The CSS statistics were computed as follows: For each constituent method, test statistics were ranked (1, …, $n$) genome-wide on $n$ SNPs. Ranks were converted to fractional ranks (r′) (between 0 and 1) by $1/(n+1)$ through $n/(n+1)$. Fractional ranks were converted to z-values as $z = \Phi-1(r')$, where $\Phi-1(\cdot)$ is the inverse normal cumulative distribution function. Mean z scores were calculated by averaging z-values across all constituent tests at each SNP position and $P$-values were directly obtained from the distribution of means from a normal $N(0, m^{-1})$ distribution where m is the number of constituent test statistics. Log-transformed $P$ values ($-\log_{10}$ of $P$ values of the mean z-values) were declared as CSS. To identify significant selection signals CSS scores were plotted against the genomic positions and the individual test statistics were then averaged across SNPs within 1 Mb sliding windows to reduce spurious signals (smoothed CSS score). Clusters of >5 SNPs among the top 1% SNPs were defined as signals of selection.

**Integration of SNPs with skeletal muscle gene expression data.** To integrate the SNP data arising from the CSS analyses with gene sets generated from functional genomics data analyses[24] we used an R software package, gwinteR[77] as follows: (1) a set of significant and non-significant SNPs (named the target SNP set) was collated across all genes in each gene set at increasing genomic intervals upstream and downstream from each gene inclusive of the coding sequence (e.g., ±10 kb, ±20 kb, ±30 kb… …±100 kb); (2) for each genomic region, a null distribution of 1000 SNP sets, each of which contains the same number of total significant and non-significant combined SNPs as the target SNP set, was generated by resampling with replacement from the search space of the total population of SNPs in the CSS SNP data set; (3) the nominal (uncorrected) CSS $P$ values for the target SNP set and the null distribution SNP sets were converted to local FDR-adjusted $P$ values ($P_{adj}$) using the fdrtool R package (version 1.2.15)[134]; (4) to test the primary hypothesis for each observed genomic interval target SNP set a permuted $P$ value ($P_{perm}$) was generated based on the proportion of permuted random SNP sets where the same or a larger number of SNPs exhibiting significant $q$-values (e.g. $q < 0.05$ or $q < 0.10$) were observed; (5) a summary output file of all SNPs in the observed target SNP set with genomic locations and $q$-values was generated for subsequent investigation.

For the Racing versus non-Racing CSS data set, there were 36,768 autosomal SNPs with nominal $P$ values that could be used for the integrative genomics analyses. For the integrative analyses of functional genomics outputs with the CSS data, two different subsets of differentially expressed genes (DEGs) in skeletal muscle were used: 1) trained rest (TR) and 2) untrained exercise (UE), representing the Thoroughbred skeletal muscle transcriptomic response to training (TR) and exercise (UE)[24]. The gene sets that were used were filtered with $P_{adj}$ <$10^{-4}$ (TR) and <$10^{-12}$ (UE) to ensure manageable computational loads, resulting in 230 (TR) and 407 (UE) input genes on autosomal chromosomes for integration with the CSS SNPs. The input gene sets can be found in Supplementary Data 5 and Supplementary Data 6.

For the Mongolian Racing versus other breeds CSS data set, there were 30,003 autosomal SNPs with nominal $P$ values that could be used for the integrative genomics analyses. The same DEG lists were used for the integrative analyses.

**Whole genome resequencing sample collection, DNA extraction, and sequencing.** Blood samples were collected from $n = 70$ individuals comprising

horses indigenous to Mongolia and China and imported horse breeds. Genomic DNA was extracted following the standard phenol-chloroform extraction procedure. For genome sequencing, a total amount of 1.5 μg of genomic DNA from each sample was used to construct a library with an insert size of ~350 bp. Paired-end sequencing libraries were constructed according to the manufacturer's instructions (Illumina Inc., San Diego, CA, USA). The DNA sample was fragmented by sonication to a size of 350 bp, then DNA fragments were end polished, A-tailed, and ligated with the full-length adapter for Illumina sequencing with further PCR amplification. PCR products were then purified, and libraries were assessed for size distribution using the Agilent 2100 Bioanalyzer and quantified using real-time PCR and sequenced on the Illumina HiSeq PE150 platform (Illumina Inc.).

**Whole genome resequencing quality control.** In total ~97 Gb raw sequence data was generated for each sample with the quality of the bases 96% and 90.92% for ≥Q20 and ≥Q30, respectively. Approximately 652 Mb clean paired-end reads with 30× coverage per horse (24–36×) were retained after removing low quality reads (Supplementary Data 12). Reads were removed with ≥10% unidentified nucleotides (N); >10 nt aligned to the adaptor, allowing ≤10% mismatches; >50% bases having phred quality <5; and putative PCR duplicates generated in the library construction process, which mainly result from base-calling duplicates and adaptor contamination.

**Whole genome resequencing reads mapping, SNP and INDEL calling.** The remaining high quality paired-end reads were mapped to the horse EquCab 3.0 reference genome using BWA (Burrows-Wheeler Aligner) (Version: 0.7.8) with the command 'mem -t 4 -k 32 –M'[135]. In order to reduce mismatch generated by PCR amplification before sequencing, duplicated reads were removed using SAMtools. After alignment, SNPs were called on a population scale using a Bayesian approach as implemented in the package SAMtools[136]. Genotype likelihoods from reads for each animal at each genomic location, and the allele frequencies in the sample were then calculated using a Bayesian approach. The 'mpileup' command was used to identify SNPs with the parameters as '-q 1 -C 50 -t SP -t DP -m 2 -F 0.002'. Then, to exclude SNP calling errors caused by incorrect mapping or indels, only high-quality SNPs (coverage depth ≥3 and ≤50, RMS mapping quality ≥20, maf ≥0.05, miss≤0.1) were retained for subsequent analysis.

SNP calling was also performed using GATK v4.1.2.0[137]. The process was as follows: (1) HaplotypeCaller generates gvcf files for each animal; gatk HaplotypeCaller -genotyping-mode DISCOVERY -ERC GVCF. (2) GATK CombineGVCFs combines all the generated gvcf files. (3) GATK GenotypeGVCFs performs the genotyping on the combined vcf file. (4) GATK SelectVariants separates the SNPs and indels. A summary of the SNPs and indels called in the 70 samples is provided in Supplementary Data 13.

GATK SelectVariants was used to select a common subset of variants between unfiltered GATK and SAMtools calling result. Approximately 2 million horse SNPs (MNEc2M) available in the public repository (www.animalgenome.org/repository/pub/UMN2018.1003) were used for variant calling validation. A summary of the validation of SNPs called in this study and MNEc2M SNPs is provided in Supplementary Data 14.

Similar to SNP calling, the calling of indels was conducted using SAMtools with minimum depth ≥3 and GQ >20, and only indels <50 bp were retained.

**Prioritisation of novel variants.** The functional effects of the variants were predicted according to the Ensembl (version 101) annotation of EquCab3. Functional consequences of the variants (including SIFT scores[138] for missense variants) were predicted with the Ensembl Variant Effect Predictor (VEP, version 91.3[139]).

These data formed the reference files that were screened for putative causative / functional variants in regions / genes of interest. Prioritisation of genes and variants was performed with a particular focus on (1) predicted effect of the variant, where 'high' effect was selected where possible, (2) biological function of the gene relevant to exercise, (3) minor allele frequency ≥0.1, and (4) high quality score (most had QUAL = 999)[140]. All genes were considered in the regions demarked by the CSS analyses, but generally only DEGs were considered in regions (±100 kb from gene start/end) demarked by the integrative analyses. Variants were only considered if they were in known genes. Predicted 'high' effect variants included the annotations splice_acceptor_variant&intron_variant, frameshift_variant, splice_acceptor_variant, and stop_gained, and predicted 'moderate' effect variants included the annotations missense_variant and inframe_deletion.

**Matrix-associated laser desorption/ionization time-of-flight mass spectrometry (MALDI-TOF MS) assay design for SNP genotyping.** Assay development and genotyping was performed at Neogen Genomics (Lincoln, Nebraska, USA) using the MassARRAY platform and iPLEX GOLD chemistry according to the manufacturer's protocol (Agena Bioscience, San Diego, California, USA). Agena Design Suite software developed by the manufacturer was utilized to design multiplex assays. The variants of interest were among a larger set of 150 markers of interest (SNPs, MNPs) (designed for other analyses not included here) that were split into four separate sets of 48, 47, 35, and 20 markers each. The four multiplex

assays were run on the genomic DNA provided for the animals, data generated, and quality check metrics applied.

**Tests of genetic association.** SNP genotype data for all samples were merged and converted to Plink .bed format. The data were analysed using PLINK 1.9[141]. SNPs with a call rate <80% and samples with a call rate <90% were excluded from the analysis. Case-control tests of genetic association were performed to compare the racing breeds horses $n = 267$ to non-Racing breeds horses $n = 249$ and to compare the Mongolian Racing horses $n = 46$ to non-Racing Chinese Mongolian breeds horses $n = 121$ (Supplementary Data 16). The Thoroughbred cohort was divided by geographic region, Europe, Australia, and North America. The quantitative trait, number of racecourse starts, and the binary trait, Elite vs non-Elite, were tested for each cohort with sex included as a co-variate (Supplementary Data 18).

**Statistics and reproducibility.** The composite selection signals (CSS) approach which combines the statistical outputs from the $F_{ST}$, ΔSAF and XP-EHH tests into one composite statistic for each SNP was used for the identification of selection signals. Significance was assigned to SNPs within clusters of >5 SNPs among the top 1% of CSS scores. The reproducibility of the results is demonstrated by the identification of several selection signals in the Thoroughbred that were previously identified using different statistical methodologies ($d_i$, H, $H_{12}$, and Tajima's D)[5,7]. Overlaps with reported selection signals in other horse breeds are shown in Table 1.

The R package gwinteR was used for the integration of SNPs with DEGs. A set of significant and non-significant SNPs from the CSS analysis was collated across all genes in the DEG gene set at increasing genomic intervals (e.g., ±10 kb, ±20 kb, ±30 kb… …±100 kb) upstream and downstream from each gene. A null distribution of 1,000 SNP sets, each of which contains the same number of total significant and non-significant combined SNPs as the target SNP set, was generated for each genomic region by resampling with replacement from the search space of the total population of SNPs in the CSS data set. The nominal (uncorrected) CSS $P$ values for the target SNP set and the null distribution SNP sets were converted to local FDR-adjusted $P$ values ($P_{adj.}$) using the fdrtool R package (version 1.2.15)[134]. To test the primary hypothesis for each observed genomic interval target SNP set a permuted $P$ value ($P_{perm.}$) was generated based on the proportion of permuted random SNP sets where the same or a larger number of SNPs exhibiting significant $q$-values (e.g. $q < 0.05$ or $q < 0.10$) were observed. A summary output file of all SNPs in the observed target SNP set with genomic locations and $q$-values was generated.

Tests of genetic association were performed by comparing allele frequencies in a chi-square (1df) test for the binary traits, and a Wald test for the quantitative trait. Sex of the horse was included as a covariate. $P$-values were adjusted for multiple testing by applying a Bonferroni correction. To demonstrate the reproducibility of the results, samples were independent of the discovery sample sets (CSS and WGS), and for the association with racing, racing breeds (Quarter Horse, French Trotter, Standardbred) that were not included in the discovery sample set were included in the racing cohort.

**Reporting summary.** Further information on research design is available in the Nature Portfolio Reporting Summary linked to this article.

## Data availability

SNP array derived genotypes generated in this study have been deposited in the European Variation Archive with the accession IDs PRJEB55561 (Project), ERZ12817059 (Mongolian horse analysis), and ERZ12817060 (Arabian horse analysis). The whole genome sequence data have been deposited in the Sequence Read Archive with the BioProject ID: PRJNA867509. The source data for Fig. 2 is available at Source data is available at https://doi.org/10.5061/dryad.g79cnp5sm. The SNP genotype data generated for the validation study are subject to the following licenses/restrictions: The phenotype and genotype data are the property of Plusvital Ltd. and subject to a confidentiality agreement with the animal owners.

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

## Acknowledgements

We thank the Khentii herdsmen and Henry Seebach for assistance with collection of the Mongolian Racing horse samples; horse owners for agreement for the use of samples in research; Jonathan O'Grady (O'Grady Advisors) for assistance compiling the Thoroughbred race record phenotypes; and Gillian McHugo for assistance with data visualisation. This work was supported by National Key R&D Program of China (Grant Code 2017YFE0108700); National Natural Science Foundation of China (Grant Code 3191101008,31960657); Science Foundation Ireland (Grant Code 19FFP6879); National Institutes of Health (Grant Code NIEHS-1D43ES02286201); Royal Agricultural University Cirencester Fund; Plusvital Ltd.

## Author contributions

H.H. compiled the SNP data, performed the population genomics analyses, coordinated the sample acquisition of Asian horses, coordinated the WGS study, interpreted the results, and assisted with writing the original manuscript. B.A.M. designed and performed the SNP association analyses, assisted with the population genomics analyses, assisted with the WGS study, interpreted the results, and assisted with writing the original manuscript. L.A. and A.J.H. were involved in planning and performing the sample acquisition and DNA extraction of Mongolian Racing horses. D.B., D.D. and T.J. were involved in the sample acquisition of Asian horses. B.J., J.D. and U.J. facilitated and were involved in the sample acquisition of Mongolian Racing horses. A.R.H. and L.R.C. prepared the phenotypes and samples for the SNP association analyses. N.K.K. and H.P. performed the downstream WGS analyses. T.J.H. and D.E.M. designed and performed the integrative genomics analyses and assisted with writing the original manuscript. C.R. and D.W. conceived the study and facilitated and performed the sample acquisition of Mongolian Racing horses. M.D. facilitated the sample acquisition of Asian horses and supervised the WGS study. E.W.H. conceived, designed and coordinated the study, performed the sample acquisition of Mongolian Racing horses, supervised the analyses, interpreted the results and wrote the original manuscript.

## Competing interests

The authors declare the following competing interests: D.E.M. and E.W.H. are shareholders in Plusvital Ltd. B.A.M., L.R.C. and A.R.H. are employees of Plusvital Ltd. Other than the authors, the funders played no role in the study design, data collection and analysis, decision to publish, or preparation of the manuscript. All other authors declare no competing interests.
