## [Peer Review File · Communications Biology]

Reviewers' comments:

Reviewer #1 (Remarks to the Author):

This manuscript describes the identification of genomic regions related to racing ability by comparing the genetic structures of three racing breeds, Thoroughbred, Arabian, Mongolian.

The authors report G6PC2, HDAC9, KTN1, MYLK2, NTM, SLC16A1 and SYNDIG1 as candidates for driver genes related to racing ability.

These finding is very interesting, and the manuscript is well written.

While this reviewer suggests accepting of the current manuscripts, the authors should revise the following minor comments.

Line 1: Is "The Sport of Kings" necessary in the title?

Line 78-80: Does this mean to Anglo-Arab?

Line 141-146: This part looks like conclusion or discussion rather than introduction?

Line 152: Are "Mongolian Racing" and "Mongolian" different breed? Mongolian Racing is a specific breed for racing in Mongolian horses? This is a closed population, such as Thoroughbred? Selected genomic regions in this study is for Mongolian Racing or General Mongolian?

Line 174: If the 5 Arabian contain thoroughbred's factors, should these horses be excluded from the following analyses? Wouldn't it be possible to get clearer results by excluding it?

Line 210: This is similar question. The results are for Mongolian Racing or General Mongolian?

Line 253 (Table 1): Why are there so few genomic regions detected in Mongolian horses? Is this due to the range of LD is shorter in Mongolia than in thoroughbreds?

Line 274 (Table 2): Should it be in chromosomal order?

Line 316 (Figure 4): Is "the genome-wide (1% SNPs) threshold" a general threshold?

Line 734: Does the difference in sample size between each population affect the results? In addition, does the difference in LD of each breed affect the result?

Line 760: The authors should describe the statistical significance (threshold) of GWAS?

Reviewer #2 (Remarks to the Author):

I am pleased to be assigned as referee of this excellent paper. In order to clarify some doubts I had while reading the manuscript, I must ask some questions and need some extra data about your work.

What was the main goal of the manuscript? I understood the aspect mentioned previously regarding the improvement of racing populations, particularly among Mongolian horses. But I think the goal of the manuscript is to identify SNPs and signaling markers related to the racing performance in these breeds.

Another important aspect mentioned in the paper is about the athletic phenotype. For me, this was not evaluated. only their athletic performance was reported.

Another crucial and confusing point was about the animals used. This data need to be more clear for the readers. Maybe a table would be more suitable to visualize which animals and/or data were used and when inside the study they were used.

Why the Belgian breed had much more attention than the other non-racing breeds? (Table S1)

In order to help me and other readers, please provide a chronogram/organogram of the steps of this study. I think it would be a great supplementary file.

You have mentioned in your text the role of some important genes related to the racing, but some

other genes reported in the supplementary files were mentioned just by their symbol and their role were never mentioned. It would of great significance to describe their role and report their full names.

Some of the tables of the supplementary data must be in the text. Review that.

The conclusion must be more assertive. It is long and unclear. That being said, I found it difficult to visualize how the population of Mongolia will use this paper as a tool to help them to improve the selection of best horses for their breeding herd.

These requests would help me a lot in understanding your work and will allow me to work with you to publish such an amazing and complex article.

With warm regards,

Felipe Gomes Ferreira Padilha.

Reviewer #3 (Remarks to the Author):

The presented study aimed to find loci across genomes of horses that potentially contribute to the galloping (racing) ability. The hypothesis was constructed on a recently "omnigenic" proposed model in which the complex trait is shaped by core genes and many genetic variants of small effect – peripheral genes located across the entire genome. The author's proof of concept was that all breeds of horses that have the ability to gallop and compete, harbor selection signals (sequence variants) potentially associated with racing performance.

To achieve aimed goals authors described the results generated using a number of methods in the field of modern molecular genetics, and run WGG on n=54 horses, WGS on n=70, and SNP's genotyping on n=516 with the use of MALDI-TOF MS assay. Furthermore, the authors integrate data from CSS analyses (mainly from data from other papers) with functional genomics data from other research.

In general, the idea of the manuscript seems interesting and the findings will improve further understanding of the genetic background of horse racing. However, certain aspects need explanation. First of all much intense attention should be paid to the classification of horse breeds and the types of utility other than TB's (Racing). If we talk about the horse's utility it is worth mentioning that racing is distinguished from endurance. Equestrian activities in theory hippology are divided into racing and sport. The sports disciplines are jumping, dressage and para-dressage, eventing, driving and para driving, endurance riding, and vaulting. So I would recommend using the proper definition of equine activities with a proper citation from authorities FEI.org for example. As above, the selection of individuals for the experiment should be more precise and samples used for analyses also. The aim of the study suggests that the main goal is to find loci that contribute to racing abilities. And authors state by themselves that recent admixture has been observed in racing Arabians and racing Mongolian horses, so why do authors generalize all populations of Arabian horses? This is now clear enough that distinct populations exist so analysis should be performed on the racing populations (as seen in the paper by Crosgrave et al. 2020) or on both. Even if collected samples are with no additional pedigree information, PCA should place them accordingly Ln 707-710). This approach will confirm that racing and endurance selection will harbor the same signals or there is different genetic background for these performances (As the endurance riding harbors more complex traits). This issue should be addressed throughout the whole manuscript.

Please find attached detailed suggestions:

The nomenclature of breeds should be consistent throughout the manuscript: Arabians (WAHO registered), Racing Arabians, Endurance Arabians, Straight Egyptian Arabians; Quarter Horse, Racing

Quarter Horse, et cetera.

Ln 80-82: Yes, but in a population without TB's ancestry.

Ln 137: If the authors would like to improve particularly the Mongolian population why do you use Arabians for analysis?

Ln 141-143: the performed analysis does not support this sentence.

Ln 210-214: this sentence needs strong literature support.

Ln 229: Table S3, and Figure 3 where detailed information is in Table S2. I would recommend to add more information to the description of the Figures. Furthermore, Signals on Eca1, 7, 17, and 18 are the highest and it is reasonable if you compare TB, and TB crosses with other breeds. And this is directly involved with flat racing rather than endurance.

Ln 260-273: The description of functional enrichment would better look if results will be divided into GO terms. Table 2, is selected randomly? This should be clarified. Table S4, information should be ordered according to e.x. GO terms.

Ln 351: The whole part of integrative genomic has been performed on RNA-Seq data from TB and the hypothesis with TB's admixture is ok. But If you also would like to refer to Arabian horses I suggest add data from RNA-Seq experiments performed on Arabians. Also, this part needs literature support according to methods.

Ln 402-404: Authors mentioned the MNEc2M array, please state why.

Ln 462-464: These breeds should be definitely corrected for example; Egyptian (?) Warmblood (some Warmbloods are more „ racing“ than „racing“ Arabians).

Ln 473: Table 4: Have authors confirmed these SNP's by Sanger Sequencing?

Ln 535: Please add a reference to support findings of SLC16A1 in Arabian racing also from the analysis of selection signatures.

Ln 604: I have the impression that it is missing the part about Genes associated with racing performance in Arabian horses.

Ln 653: On what criterion did the authors classify the horses as Arabians?

We would like to thank the reviewers for their helpful comments. We have responded below to each comment point by point.

Kind regards, Emmeline

Referee expertise:

Reviewers' comments:

Reviewer #1 (Remarks to the Author):

This manuscript describes the identification of genomic regions related to racing ability by comparing the genetic structures of three racing breeds, Thoroughbred, Arabian, Mongolian. The authors report G6PC2, HDAC9, KTN1, MYLK2, NTM, SLC16A1 and SYNDIG1 as candidates for driver genes related to racing ability.

These finding is very interesting, and the manuscript is well written.

While this reviewer suggests accepting of the current manuscripts, the authors should revise the following minor comments.

Reviewer comment	Response
Line 1: Is "The Sport of Kings" necessary in the title?	We have removed this from the title
Line 78-80: Does this mean to Anglo-Arab?	This refers to the 'flat racing Arabian' horses used in the Cosgrove et al 2020 paper. We reference this paper directly after that sentence to indicate the population referred to: Cosgrove methods section describes them as follows: " A set of flat racing Arabian samples was identified as the putatively admixed group. This "Racing Arabian" group (N = 34) included all horses known to compete in flat racing, as well as five additional samples with unknown performance records that clustered with the flat racing samples in PCA (Fig. 2; five Arabian samples with PC1 < -40 and PC2 > 20)."
Line 141-146: This part looks like conclusion or discussion rather than introduction?	We have deleted this part
Line 152: Are "Mongolian Racing" and "Mongolian" different breed? Mongolian Racing is a specific breed for racing in Mongolian horses? This is a closed population, such as Thoroughbred? Selected genomic regions in this study is for Mongolian Racing or General Mongolian?	Mongolian Racing is a subtype of Mongolian horse used for racing in Mongolia. It is not a closed population. Selected genomic regions are for Mongolian Racing as noted in Introduction (line 132), results (line 287), and methods (line 648).
Line 174: If the 5 Arabian contain thoroughbred's factors, should these horses be excluded from the following	This is a population based approach, and therefore it is expected that the genomes of the population as an entirety represent the genetic variation in the

analyses? Wouldn't it be possible to get clearer results by excluding it?	population. In our PCA of Arabian horses (Figure S1) the horses were as broadly distributed in PC2 as the Mongolian Racing horses were in the PCA plot for Mongolian horses (Figure S2) and therefore we believe it is appropriate that they remain in the population analysis.
Line 210: This is similar question. The results are for Mongolian Racing or General Mongolian?	Mongolian Racing
Line 253 (Table 1): Why are there so few genomic regions detected in Mongolian horses? Is this due to the range of LD is shorter in Mongolia than in thoroughbreds?	Table 1 shows selection signals identified when all Racing breeds were analysed together (columns to the left). The columns on the right indicate if the same selected region was also detected when the individual racing populations were used as the target selected population on their own. Table S3 shows all the selected regions detected in Mongolian Racing, and in fact there are more detected regions in Mongolian Racing than when all the Racing breeds are included together.
Line 274 (Table 2): Should it be in chromosomal order?	Table 2 is ordered by ranking the strongest selected region (based on CSS score) to the least strong, which we believe is the most appropriate way to illustrate the results. Table S3 is also organised in this way.
Line 316 (Figure 4): Is “the genome-wide (1% SNPs) threshold” a general threshold?	Yes, this is standard. 1% or 0.1% SNPs may be used. We followed the procedure of Randhawa et al (2014). Randhawa et al Methods section notes: “In the absence of a known probability distribution for most cases of the test statistics used in this study, SNPs with extreme test scores (top 0.1%) in the genome-wide distribution were considered significant (Kijas et al 2012)” (Randhawa et al 2014 Composite selection signals can localize the trait specific genomic regions in multi-breed populations of cattle and sheep. BMC Genetics) We have previously published using this method (Han et al 2020 Plos One Selection in Australian Thoroughbred horses acts on a locus associated with early two-year old speed). See also below excerpts for examples from recently published papers “We plotted the window-averaged di values across the chromosome, and the top 1% windows of the empirical distribution were defined as putative selection regions.” Capturing Genetic Diversity and Selection Signatures of

	the Endangered Kosovar Balusha Sheep Breed. Adeniyi OO, Simon R, Bytyqi H, Kugler W, Mehmeti H, Berisha K, Simčić M, Magdy M, Lühken G. Genes (Basel). 2022 May 12;13(5):866. doi: 10.3390/genes13050866. “To identify the genomic regions of “high homozygosity,” also called ROH islands, the top 0.999 SNPs of the percentile distribution of the locus homozygosity range within each breed were selected”. A Combined Multi-Cohort Approach Reveals Novel and Known Genome-Wide Selection Signatures for Wool Traits in Merino and Merino-Derived Sheep Breeds. Megdiche S, Mastrangelo S, Ben Hamouda M, Lenstra JA, Ciani E. Front Genet. 2019 Oct 25;10:1025. doi: 10.3389/fgene.2019.01025. eCollection 2019.
Line 734: Does the difference in sample size between each population affect the results? In addition, does the difference in LD of each breed affect the result?	It is common in these types of studies to have variable sample sizes per group and with different breed population demographic histories (that may influence LD). This does not affect the power to detect selection regions as evidenced in Randhawa et al 2014 Table 1 which shows the results for known major effect genes for a range of traits identified using CSS. For example, for double muscling trait, n=149 (double muscling) were compared to n=2654 (normal muscling) and the MSTN gene was identified. Other examples are also shown in that table. Randhawa et al 2014 notes: “The comparison of each test by including and excluding breeds with less than 10 and 20 animals showed negligible differences for the effect of variable sample size (especially low) of breeds in European group (Additional file 14: Figure S9). It shows that breeds with a similar history generally have shared patterns of genetic diversity. In addition, it also provides evidence that computation of CSS is not sensitive to the individual sample size of the participating breeds for outbred populations.”
Line 760: The authors should describe the statistical significance (threshold) of GWAS?	Thank you, we have added the threshold to the methods section (line 654). For clarity, we think the reviewer must be referring to the CSS analysis (not a GWAS).

Reviewer #2 (Remarks to the Author):

I am pleased to be assigned as referee of this excellent paper. In order to clarify some doubts I had while reading the manuscript, I must ask some questions and need some extra data about your work.

Reviewer comment	Response
What was the main goal of the manuscript? I understood the aspect mentioned previously regarding the improvement of racing populations, particularly among Mongolian horses. But I think the goal of the manuscript is to identify SNPs and signaling markers related to the racing performance in these breeds.	Thank you, this has been clarified at the end of the Introduction.
Another important aspect mentioned in the paper is about the athletic phenotype. For me, this was not evaluated. only their athletic performance was reported.	We refer to phenotype in the broad sense - breeds that are selected on the basis of racing performance are expected to have an athletic / racing phenotype (rather than breeds used for riding, leisure, or other purposes). We have clarified this by editing the text and referring to the racing phenotype (rather than athletic phenotype or racing athletic phenotype, which we realise was confusing). Indeed, further to this suggestion we have changed the title of the paper to reflect this. Thank you for the suggestion.
Another crucial and confusing point was about the animals used. This data need to be more clear for the readers. Maybe a table would be more suitable to visualize which animals and/or data were used and when inside the study they were used.	This information is provided for CSS in Table S2, for whole genome sequence in Table S11, and for validation genotyping in Table S16.
Why the Belgian breed had much more attention than the other non-racing breeds? (Table S1)	This was not intentional, we have revised and moved BEL to the end of the table.
In order to help me and other readers, please provide a chronogram/organogram of the steps of this study. I think it would be a great supplementary file.	Thank you for the suggestion, we have added a new flow chart to the Supplementary Information document
You have mentioned in your text the role of some important genes related to the racing, but some other genes reported in the supplementary files were mentioned just by their symbol and their role	Thank you for noticing this. We have added the official gene name to the Suppl. Information document, where the functions of the genes chosen for validation genotyping are described.

were never mentioned. It would of great significance to describe their role and report their full names.	
Some of the tables of the supplementary data must be in the text. Review that.	All of the supplementary tables and figures are referred to throughout the text document. I have checked that all of them are noted in the text.
The conclusion must be more assertive. It is long and unclear. That being said, I found it difficult to visualize how the population of Mongolia will use this paper as a tool to help them to improve the selection of best horses for their breeding herd.	Thank you, we have shortened the conclusion, which we realise was more of a summary of the work done rather than a conclusion. Some Mongolian horse breeders (including the collaborators on the paper) are interested in using genetic selection to develop their herds. As well as within Mongolia, there is considerable effort in China to refine the racing populations of Mongolian horses, which is being led by IMAU researchers and collaborators.

These requests would help me a lot in understanding your work and will allow me to work with you to publish such an amazing and complex article.

With warm regards,

Felipe Gomes Ferreira Padilha.

Reviewer #3 (Remarks to the Author):

The presented study aimed to find loci across genomes of horses that potentially contribute to the galloping (racing) ability. The hypothesis was constructed on a recently "omnigenic" proposed model in which the complex trait is shaped by core genes and many genetic variants of small effect – peripheral genes located across the entire genome. The author's proof of concept was that all breeds of horses that have the ability to gallop and compete, harbor selection signals (sequence variants) potentially associated with racing performance.

To achieve aimed goals authors described the results generated using a number of methods in the field of modern molecular genetics, and run WGG on n=54 horses, WGS on n=70, and SNP's genotyping on n=516 with the use of MALDI-TOF MS assay. Furthermore, the authors integrate data from CSS analyses (mainly from data from other papers) with functional genomics data from other research.

In general, the idea of the manuscript seems interesting and the findings will improve further understanding of the genetic background of horse racing. However, certain aspects need explanation. First of all much intense attention should be paid to the classification of horse breeds and the types of utility other than TB's (Racing). If we talk about the horse's utility it is worth mentioning that racing is distinguished from endurance. Equestrian activities in theory hippology are divided into racing and sport. The sports disciplines are jumping, dressage and para-dressage, eventing, driving and para driving, endurance riding, and vaulting. So I would recommend using the proper definition of equine activities with a proper citation from authorities FEI.org for example. As above, the selection of individuals for the experiment should be more precise and samples used for analyses also. The aim of the study suggests that the main goal is to find loci that contribute to racing

abilities. And authors state by themselves that recent admixture has been observed in racing Arabians and racing Mongolian horses, so why do authors generalize all populations of Arabian horses? This is now clear enough that distinct populations exist so analysis should be performed on the racing populations (as seen in the paper by Cosgrove et al. 2020) or on both. Even if collected samples are with no additional pedigree information, PCA should place them accordingly Ln 707-710). This approach will confirm that racing and endurance selection will harbor the same signals or there is different genetic background for these performances (As the endurance riding harbors more complex traits). This issue should be addressed throughout the whole manuscript. g

Please find attached detailed suggestions:

The nomenclature of breeds should be consistent throughout the manuscript: Arabians (WAHO registered), Racing Arabians, Endurance Arabians, Straight Egyptian Arabians; Quarter Horse, Racing Quarter Horse, et cetera.

Reviewer comment	Response
See above	Thank you for your helpful comments. To confirm the type of Arabian horse used in the study we have performed a PCA for our samples and using genotypes from Cosgrove et al for the horses with recorded performance use (show, racing, endurance). We have included the PCA in the Supplementary Information file as Figure S6 (it is also at the end of this document). The horses genotyped and used for the selection signals analysis fall within the distribution of variation shared predominantly with racing Arabians. Therefore, we are confident that including this sample will improve the identification of selection signals for racing. We have added more details to the Methods section (Populations and samples) and to the Supplementary Information to explain this and to more clearly define the sample used.
Ln 80-82: Yes, but in a population without TB's ancestry.	I'm sorry I didn't understand this question
Ln 137: If the authors would like to improve particularly the Mongolian population why do you use Arabians for analysis?	The purpose was to use the power of several racing breeds to identify racing-specific selection signals. Therefore, we included the Arabian population to improve the chance to identify racing genes. Randhawa et al 2014 states "detection of signatures of strong selection can be boosted by combining samples from multiple breeds based on known traits and compare such multi-breed populations for the contrasting phenotypes".
Ln 141-143: the performed analysis does not support this sentence.	We have deleted lines 140-145
Ln 210-214: this sentence needs strong literature support.	We have added a sentence explaining how the CSS approach uses several component statistics such that only the strongest signals of selection are identified common to all three constituent tests and therefore the recent admixture in small numbers of individuals is unlikely to affect results.

Ln 229: Table S3, and Figure 3 where detailed information is in Table S2. I would recommend to add more information to the description of the Figures.	We have added more detail to the figure legend to summarise the phenotypes defining the non-racing breeds.
Furthermore, Signals on Eca1, 7, 17, and 18 are the highest and it is reasonable if you compare TB, and TB crosses with other breeds. And this is directly involved with flat racing rather than endurance.	Following your helpful comments, during the review process we performed the same selection signals analysis for the 'endurance Arabians' from Cosgrove et al. See Table Rev1 at the end of this document. The top signal for Endurance Arabian was on ECA7, which was the 2nd ranked region for 'Racing' that contains the NTM gene. Four of the Endurance Arabian selection signals directly overlapped with the selection signals that we obtained for our sample of Arabian racing horses. Since the purpose of this study was not to examine selection signals for Arabian horses, per se, we have not included this additional analysis for Endurance Arabians in the manuscript. However, the table of results is included at the end of this document for your own interest.
Ln 260-273: The description of functional enrichment would better look if results will be divided into GO terms. Table 2, is selected randomly? This should be clarified. Table S4, information should be ordered according to e.x. GO terms.	For clarification, we have used italicized font to indicate the GO terms and added the GO reference numbers. Table 2 is not selected randomly, these were the GO Biological Processes that were most relevant to exercise and contained genes in more than one selected regions (i.e. they were not clustered in a single selection signal, unlike for genes assigned to the gamma-aminobutyric acid signalling pathway and GABAergic synapse GO terms) Table S4 is ordered according to P-value, which we believe is the most appropriate way to show the most important pathways enriched for racing.
Ln 351: The whole part of integrative genomic has been performed on RNA-Seq data from TB and the hypothesis with TB's admixture is ok. But If you also would like to refer to Arabian horses I suggest ad data from RNA-Seq experiments performed on Arabians. Also, this part needs literature support according to methods.	We did not perform the integrative analysis for the Arabian racing horses at all (i.e. we did not use TB RNA-seq for Arabians). The reason for this was we used the Arabians for the sole purpose to strengthen the CSS analysis. We had no additional samples from Arabian horses with racing performance phenotypes that would allow us to validate the genes / SNPs – unlike for TB and Mongolian Racing. Therefore, there is no need to use RNA-seq data from Arabians for analysis for this paper. However, we agree that it would be very interesting additional work for another paper focusing on Arabian horses. The reference to the methods used is included in the manuscript text "The R software package gwinter (Hall et al. 2020) was used to determine whether genomic regions containing SNPs that are proximal to genes within the DEG sets were enriched for significance in the CSS analysis for Racing versus non-Racing breeds." Here is the link to the paper:

	https://bmcgenomics.biomedcentral.com/articles/10.1186/s12864-021-07643-w Hall T.J., Mullen M.P., McHugo G.P., Killick K.E., Ring S.C., Berry D.P., Correia C.N., Browne J.A., Gordon S.V. & MacHugh D.E. (2020) Integrative genomics of the mammalian alveolar macrophage response to intracellular mycobacteria. bioRxiv 2020.08.25.266668.
Ln 402-404: Authors mentioned the MNEc2M array, please state why.	We used the SNPs on the MNEc2M array to validate the SNP calling process from whole genome resequencing data of 70 Asian horses. As the genotyping concordance is high, it means our SNP calling is reliable. We have clarified this in the text.
Ln 462-464: These breeds should be definitely corrected for example; Egyptian (?) Warmblood (some Warmbloods are more „racing” than „racing” Arabians).	We have clarified this in the text
Ln 473: Table 4: Have authors confirmed these SNP’s by Sanger Sequencing?	No, the SNPs went through a process of QC during the design phase and following genotyping.
Ln 535: Please add a reference to support findings of SLC16A1 in Arabian racing also from the analysis of selection signatures.	I’m sorry, I don’t know which paper you are referring to? I am unable to find a reference showing the identification of SLC16A1 in selection signals for Arabian racing.
Ln 604: I have the impression that it is missing the part about Genes associated with racing performance in Arabian horses.	We did not have additional samples with racing phenotype information for Arabian horses to perform validation association analyses, so we could only do this for Tb and Mongolian Racing.
Ln 653: On what criterion did the authors classify the horses as Arabians?	We had no additional information other than the owner’s classification of the horses as Arabian racing horses. We have included additional analyses (PCA) demonstrating that the horses are most similar to Racing Arabians.

Figure S6: PCA plot for 182 Arabian horses colour coded by competition use. Arabian (ARR) horses genotyped in this study (green) are predominantly distributed in the PCA space that includes the Racing horses (orange).

Table Rev1: Selection signals identified in Endurance Arabian (from Cosgrove et al 2020) versus other breeds. Selection signals that overlapped with our Arabian selection signals (ARR, racing) generated in this study and included in the manuscript are indicated.

Chr	Cluster SNPs (N)	Cluster region (Mb)	CSS max score in cluster	Cohort	Genes in region (all 500kb up/downstream flanking SNPs)	Overlap with Racing Arabian selection signal
chr7	16	40.56-42.2	6.090977	Endurance Arabian	ZBTB44, ST14, SPATA19, SNX19, PRDM10, OPCML, NTM, NFRKB, IGSF9B, APLP2, ADAMTS8, ADAMTS15	yes
chr11	16	29.9-31.25	5.691486	Endurance Arabian	TRIM25, TOM1L1, TMEM100, STXBP4, SCPEP1, NOG, MSI2, MMD, KIF2B, HLF, DGKE, COX11, COIL, ANKFN1, AKAP1	
chr22	8	31.83-32.35	5.486769	Endurance Arabian	ZHX3, TOP1, PTPRT, PLCG1, MAFB, LPIN3, EMILIN3, CHD6	
chr22	7	16.62-17.37	5.353056	Endurance Arabian	TRMT6, TMX4, SHLD1, PROKR2, PLCB1, MCM8, LRRN4, HAO1, GPCPD1, FERMT1, CRLS1, CHGB, CDS2, BMP2	yes
chr5	6	62.3-62.72	4.958248	Endurance Arabian	VCAM1, TRMT13, SLC35A3, SLC30A7, SASS6, S1PR1, RTCA, PLPPR5, PLPPR4, PALMD, OLFM3, MFSD14A, LRRC39, GPR88, FRRS1, EXTL2, DPH5, DBT, CDC14A, AGL	
chr2	10	81.63-82.21	4.914095	Endurance Arabian	TRIM2, TMEM154, TIGD4, SH3D19, RPS3A, MAB21L2, LRBA, GATB, FHDC1, FBXW7, FAM160A1, ARFIP1	
chr3	12	32.73-33.43	4.229453	Endurance Arabian	ZDHHC7, WFDC1, USP10, TLDC1, TAF1C, SLC38A8, OSGIN1, NECAB2, MTHFSD, MLYCD, MBTPS1, KLHL36, KIAA0513, KCNG4, IRF8, HSDL1, HSBP1, GSE1, GINS2, FOXL1, FOXF1, FOXC2, FAM92B, EMC8, DNAAF1, CRISPLD2, COX4I1, COTL1, CDH13, ATP2C2, ADAD2	yes
chr14	10	85.04-85.51	4.135275	Endurance Arabian	ZFYVE16, ZCCHC9, THBS4, TENT2, SSBP2, SPZ1, SERINC5, RPS23, RASGRF2, MTX3, MSH3, HOMER1, FAM151B, DHFR, CMYA5, CKMT2, ATP6AP1L, ATG10, ANKRD34B, ACOT12	
chr7	5	34.94-35.02	4.099599	Endurance Arabian	VSIG2, VSIG10L2, TMEM218, TBRG1, STT3A, SPA17, SLC37A2, SIAE, ROBO4, ROBO3, PUS3, PKNOX2, PATE3, PATE2, PATE1, PANX3, NRG1, MSANTD2, HYLS1, HEPN1,	

					HEPACAM, FEZ1, ESAM, EI24, DDX25, CHEK1, CDON, CCDC15, ACRV1	
chr1	6	76.39-77.14	4.085190	Endurance Arabian	TOMM20, TBCE, TARBP1, SLC35F3, RYR2, RBM34, NID1, MTR, LYST, LGALS8, IRF2BP2, HEATR1, GPR137B, GNG4, GGPS1, ERO1B, EDARADD, COA6, B3GALNT2, ARID4B, ACTN2	
chr7	5	97.83-97.85	3.571321	Endurance Arabian	MPPED2, METTL15, KCNA4, FSHB, ARL14EP	
chr2	6	101.03-101.3	3.419062	Endurance Arabian	SLC25A31, SCLT1, PLK4, MFSD8, LARP1B, JADE1, INTU, HSPA4L, ABHD18	yes
chr21	5	43.74-43.76	3.285404	Endurance Arabian	CDH18, BASP1	

REVIEWERS' COMMENTS:

Reviewer #1 (Remarks to the Author):

The resubmitted manuscript was well revised. Well Done!

Reviewer #2 (Remarks to the Author):

The study aimed to identify which genes are prone to interfere with the racing performance in different breeds of racehorses.

With the changes made, the ideas were clarified and it is more approachable to the academic society and general public.

I recommend the acceptance of the manuscript under minor revision.

Conclusion: The genes found to be key drivers of the racing phenotype must be addressed. Lines 607 - 612 must be the last paragraph of the "results and discussion".

Best regards,
Felipe Gomes Ferreira Padilha.

Reviewer #3 (Remarks to the Author):

Dear authors, thank you for considering all my comments and suggestion. Overall I have just one impression. There is just one definition of Arabian horse accepted with WAHO. Thus all horses which do not fit within the definition are not Arabian horses. So if your samples are not from individuals registered in WAHO accepted studbook, you should not call them Arabians (629-632). You might use in Arabian type for example. Furthermore, it is similar to TB, it is not important if horses are from New Zeland USA or Australia, You still call them TB, If are registered by authorities. If not, end up as warmblood.

I understand it might be quite confusing, however in my opinion we should do science as part of the practice.

The table for Rec1 is also so nice 😊.

Reviewer #2:

Reviewer Comment	Response
Conclusion: The genes found to be key drivers of the racing phenotype must be addressed.	We have included the list of genes in the conclusion
Lines 607 - 612 must be the last paragraph of the "results and discussion".	Sorry, I disagree with this. As lines 607-612 are written as a summary statement, they should be included in the Conclusion. All of the information is already included in various sections of 'results and discussion'.

Reviewer #3:

Reviewer Comment	Response
There is just one definition of Arabian horse accepted with WAHO. Thus all horses which do not fit within the definition are not Arabian horses. So if your samples are not from individuals registered in WAHO accepted studbook, you should not call them Arabians (629-632). You might use in Arabian type for example. Furthermore, it is similar to TB, it is not important if horses are from New Zeland USA or Australia, You still call them TB, If are registered by authorities. If not, end up as warmblood.	Thank you for noting the definition of Arabian horse according to WAHO. I have read the detailed information on the WAHO website: Arabian Horse Definition 2015 (waho.org) The WAHO Arabian Horse Definition reads as follows: "A Purebred Arabian horse is one which appears in any purebred Arabian Stud Book or Register listed by WAHO as acceptable." To clarify, we have addressed this further in the Supplemental Information (Supplementary Note 1) by including the following sentences: The Arabian samples used in this study were submitted for genetic testing from the United Arab Emirates. No additional pedigree information was available. Therefore, we cannot be certain that the horses fulfil the requirement of the World Arabian Horse Organisation (WAHO) for the definition of a Purebred Arabian that states: "A Purebred Arabian horse is one which appears in any purebred Arabian Stud Book or Register listed by WAHO as acceptable." Since the samples fall within the genetic variation for other Arabian horses in the public domain, throughout the manuscript the population is referred to simply as Arabian, as it is uncertain that they fulfil the definition of 'Purebred Arabian'.